



# Comparative analysis and implications of sustainable Flood Risk Management in four front-end countries: The United Kingdom, the Netherlands, the United States, & Japan

Faith Ka Shun Chan[1,2*], Liang Emlyn Yang[3*], Gordon Mitchell[4], Nigel Wright[5], Mingfu Guan[6], Xiaohui
Lu[7*], Zilin Wang[8], Burrell Montz[9] and Olalekan Adekola[10]

[1]School of Geographical Sciences, University of Nottingham Ningbo Campus, Ningbo 315100, China; email:
faith.chan@nottingham.edu.cn
[2]Water@Leeds Research Centre, University of Leeds, Leeds LS29JT, UK, email: k.s.chan@leeds.ac.uk
[3]Department of Geography, Ludwig Maximilian University of Munich (LMU), email: emlyn.yang@lmu.de
[4]School of Geography and Water@Leeds Research Centre, University of Leeds, Leeds LS29JT, UK, email:
g.mitchell@leeds.ac.uk
[5]Nottingham Trent University, 50 Shakespeare Street, Nottingham, Nottinghamshire, United Kingdom, email:
nigel.wright@ntu.ac.uk
[6]Department of Civil Engineering, The University of Hong Kong, Pok Fu Lam, Hong Kong, email: mfguan@hku.hk
[7]Key Lab of Urban Environment and Health, Institute of Urban Environment, Chinese Academy of Sciences, Xiamen 361021,
China & Nottingham University Business School China, University of Nottingham Ningbo China, Ningbo 315100, China,
email: xhlu@iue.ac.cn/xiaohui.lu@nottingham.edu.cn
[8]School of Geographical Sciences, University of Nottingham Ningbo Campus, Ningbo 315100, China; email:
wangzilin.0726@gmail.com
[9]Environmental Geography and Planning, email: montzb@ecu.edu
[10]School of Humanities, York St John University, email: o.adekola@yorksj.ac.uk

*Correspondence to*: Faith Ka Shun Chan (faith.chan@nottingham.edu.cn/k.s.chan@leeds.ac.uk); Liang Emlyn Yang
(emlyn.yang@lmu.de); Xiaohui Lu (xiaohui.lu@nottingham.edu.cn/xhlu@iue.ac.cn)

**Abstract.** Sustainable flood risk management (SFRM) has become popular since the 1980s. Many governmental and non-governmental organisations have been keen on implementing the SFRM strategies by integrating social, ecological and economic themes into their flood risk management (FRM) practices. However, justifications for SFRM are still embryonic and it is not yet clear whether this concept is influencing the current policies in different countries. This paper reviews the past and present flood management approaches and experiences from flood defence to FRM in four developed countries with the

aim of highlighting lessons for developing mega deltas. The paper explored recent strategies such as "*Making Space for Water, PPS 25, and NPPF*" in the UK; "*Room for Rivers*" in the Netherlands which was promoted to cope with flooding, integrate FRM with ideas on sustainability, and deliver good FRM practice for next generations. The United States has also established a sound National Flood Insurance Program (NFIP), and Japan has developed an advanced flood warning and evacuation contingency system to prepare for climatic extremes. These case studies showed some good lessons to achieve long term

SFRM direction to deliver flood management practices with social-economic and environmental concerns. Most of developing coastal megacities especially in Asia are still heavily reliant on traditional hard-engineering approach, that may not be enough



to mitigate substantial risks due to human (exist huge populations, rapid socio-economic growth, subsidence) and natural (climate change) factors. We understand different countries and cities have their own interpretation on SFRM, but recommend policy makers to adopt "mixed options" towards thinking about long term and sustainability that with social, economic and
environmental considerations.

**Keywords:** Coastal megacities, Flood risk, flood management paradigms, Sustainable Development.

## 1 Introduction

Flooding is a natural phenomenon which occurs from time to time over the long human history (Plate, 2002; Yang, et al., 2019). Floods can be caused by an intense or a long precipitation event, wave surges or exceptionally high tide events or
combined with surge and tidal events (from cyclonic monsoon effects), or by rapid snowmelt (Kundzewicz, 1999). In Asian coastal megacities, flood risk is especially high and still rising (Hanson et al., 2011; Hallegatte et al., 2013; Yang, et al. 2015; Chan, et al., 2021), through two routes. First, flood hazard is increasing as climate change raises sea surface temperature, driving oceanic thermal expansion and increases the intensity and frequency of precipitation events (Meehl et al., 2007; Hulme et al., 2002). Second, the assets exposed to flood hazards, including people and property, are growing through rapid land use
change, urbanisation and economic expansion (Yang, et al., 2010). Ways to mitigate flood risk in urban coastal areas are varied, and in principle could necessitate relocation of people and properties from high risk areas. However, such measures are extreme and costly, and are likely seen as impractical, with few people and firms willing to relocate, particularly from within dense economically vibrant cities such as Guangzhou, Shanghai, Jakarta, Ho Chi Minh City, Bangkok and Singapore. A major challenge for such cities is then to manage flood risks, whilst addressing development needs.
Looking elsewhere along the Asian coastal cities, for example, cyclone Nargis flooded Yangon, Myanmar in 2008 and led to more than 140,000 casualties and US$17 billion in economic impacts (Terry et al., 2012). In 2011, the Chao Phraya River catchment flood caused severe inundation in Bangkok, Thailand, flooded several districts of the city and caused serious economic losses, exceeding US$4 billion (Chan et al., 2018). These examples demonstrate that Asian coastal cities are exposed to several types or combined types of floods (e.g. surface water/waterlogging, fluvial in urban catchments, and coastal, etc.),
and the impacts and consequences are highly related to continuous growing economies and populations. In addition, these cities are also experiencing complex human-induced factors (e.g. reclamation of coastal areas without considering sea-level rise and surges, inadequate urban drainage system, and over-extraction of groundwater resources that enhances land subsidence, to name a few). Experience from other countries which have faced severe flooding suggests lessons for dealing with the flood risk. Asian megacities tend to rely on a one-dimensional, hard engineering approach to protect against flooding
(Chan et al., 2018; Yang et al., 2020), whereas in many other places, this approach is increasingly seen as untenable, as it is financially unrealistic to protect against all floods.

Therefore, flood risk management (FRM) has developed, an approach that addresses not just structural defence, but preparation (e.g. land use zoning, adaptation), non-structural protective measures, population preparedness, and emergency response and



recovery mechanisms to reduce flood risk (Samuels, 2006). FRM reflects a growing awareness of the uncertainties,
vulnerabilities and costs associated with flooding, and is the flood paradigm widely accepted in Europe and some other
advanced economies (Janssen, 2008), but to a much lesser extent in Asian cities. With the intention of drawing lessons for
Asian coastal megacities, in this paper we review FRM experience in four economies which are at the front end in applying
the FRM approach (the UK, NL, United States and Japan) (section 2). It additionally explores the wider concept of Sustainable
FRM that extends the scope of FRM to consider wider social and environmental goals (section 3). This paper also reviews the
development of flood management practices in five selected Asian coastal cities (Jakarta, Indonesia; Bangkok, Thailand; Ho
Chi Minh City, Vietnam; Guangzhou, China and Shanghai, China) to demonstrate the latest progress of FRM in these cities
that are exposed to flooding from various sources and to understand the implications of SFRM elsewhere for effectively
influencing their flood management practices.

Our overall aim of the paper is to contribute to the understanding of SFRM and practice, and to argue for consideration of
flood coping strategies for Asian coastal megacities developed through a consideration of sustainable development principles.
Making such information available is of prime importance due to the fact that Asian megacities already face massive flood
management challenges, and flood damages – whether human or material – are growing rapidly.

## 2 Learning from the four front-end countries

### 2.1 The road from traditional flood management to SFRM

Responses to flooding have historically developed through several distinct phases. Initially (pre-1980's) responses were aimed
at controlling and defending against floods by relying on "*hard engineering solutions*" (Penning-Rowsell et al., 2006). This
paradigm of "*flood control*" (1950s to 1960s) sought to reduce flooding that might damage agricultural production and
compromise food security. Land drainage constructions and channelization were used to quickly drain and keep flood water
away from agricultural lands and avoid flood impacts on agriculture. Afterwards, the paradigm of "*flood defence*" (1970s) was
adopted, as the economic interest of flood control was widened to encompass manufacturing and tertiary industries. This "*flood
defence*" phase sought to protect infrastructure, people, and their property by using structural engineering measures such as
seawalls, dykes, embankments, breakwaters and levees.

Through the 1980s a "*flood management*" approach emerged, which emphasised coping with flooding rather than solely
controlling floods. This occurred as policy makers realised it was increasingly difficult to defend against all flooding due to
increasing climatic extremes (i.e. intensive precipitation), whilst social and environmental concerns were also recognised as
important. This change in focus gave greater attention to flood preparedness and public awareness through, for example, the
flood forecasting and warning systems that were developed across Europe. Such changes have been the basis of further
development of the FRM paradigm (Lumbroso et al., 2011; Parker and Fordham, 1996).  In the 2000s, the focus changed again
to more explicitly consider flood risk (Plate, 2002), including the probability of a given flood hazard (e.g. precipitation, storm
frequency, sea-level rise) (Kundzewicz et al., 2002; Schanze et al., 2005; Tol et al., 2003) and the vulnerability of, and



consequences for populations and economic assets exposed to that flood hazard (Brown and Damery, 2002; Schanze, 2006). Thus, FRM now seeks to prevent damage by reducing the exposure and vulnerability of people and properties prone to flooding. It is not possible to eliminate flood risk, hence FRM considers the costs and benefits of flood risk mitigation for society at large (Butler and Pidgeon, 2011). The objective of FRM is thus to reduce the harmful consequences of flooding and

to balance risk reduction alongside other political considerations and priorities. An important aspect of FRM that is encouraging to manage flood risk with wider stakeholders (e.g. households, practitioners, politicians, flood engineers, planners, and communities), so as to identify multi-disciplinary perspectives and solutions (Pitt, 2008, Yang, 2020).

In this context, it is useful to reflect on national experiences of FRM with a view to identifying lessons for countries where flood defence and control continue to be the dominant response to flooding. Therefore, we next consider experience of flood

management in the UK, NL, US and Japan, where flood management practice has evolved significantly in recent decades, and show how practice has evolved from defending against flood to living with floods. These experiences offer lessons for FRM in Asian coastal megacities.

## 2.2 The United Kingdom

From the early part of the twentieth century, clear phases of flood management history can be identified in the UK, with the

first phase pre-1970s. During this period, flood policy was governed and implemented by the Ministry of Agriculture, Fisheries and Food (MAFF), whilst the Internal Drainage Board (IDB) was responsible for carrying out flood alleviation practices in the low-lying parts of UK. The foci were on land drainage and hard engineered defences such as river straightening, construction of embankments and levees. The failure of this approach became evident following major floods in 1947 and 1953 which inundated 65,000 ha and 280,000 ha of farmland respectively, and which damaged agricultural output during the post war

period when many foodstuffs were still rationed (Penning-Rowsell and Chatterton, 1977).

During the 1970s, the government adopted cost-benefit appraisal (CBA) to evaluate drainage projects. Where MAFF found high-yielding crops (particularly cereal, sugar and potatoes) to be threatened, the IDB and local authorities would help farmers to develop measures to control flooding, usually through construction of levees and drainage channels (Penning-Rowsell and Green, 2000). Johnson et al. (2007) argued that despite the application of CBA by government, flood policy was biased in

favour of farmers and land owners, as they were the major (private) beneficiaries of public expenditure. However, the UK remained unusual in that it was one of few countries at the time to apply CBA to flood control measures.

During the 1980s, the priority of UK flood policy was to "*keep the water out*" (Johnson et al., 2007). The emphasis had moved from protecting farmland, to protecting a broader asset base that underpinned economic growth. Thus, policy makers were keen on implementing flood alleviation schemes and projects that defended people and property (homes and businesses)

(Johnson and Priest, 2008). Criticism was however now directed at the use of CBA, for being overly focused on economic aspects. For example, flood measures/schemes tended to be approved if they protected high value properties in a floodplain (e.g. riverside houses) but ignored adverse ecological effects (e.g. fish and invertebrates affected by channelisation) and other environmental impacts caused by the flood defence projects themselves (Green et al., 1991; Hey et al., 1994; Penning-Rowsell





et al. 2006). Thus, through the 1980s and 1990s, flood defence was largely driven by CBA that considered net benefits (but
not their distribution) and neglected externality effects.

In England and Wales, more than 4 million people, and property valued at more than £200 billion are currently located in areas
at risk of a 1-in-100-year flood (Lo and Chan, 2017). Forecast flood damages are currently £1.4 billion annually, but are
expected to rise to as much as £27 billion by 2100 (Evans et al., 2006). In 2002, the Institute of Civil Engineers emphasised
that flood engineering measures remain important, but will no longer be enough, and advocated the approach of "*living with*
*floods*" (Fleming, 2002). This gave impetus to a further paradigm shift in flood management policy, with concern for wider
aspects, recognising socio-economic and environmental values, and impacts of climate change (Hall et al., 2003). Innovations
in flood and coastal erosion risk management included the "*Making Space for Water*" (MSW) strategy (Defra, 2007) and
"*Planning Policy Statement 25*" (PPS25) (DCLG, 2007) which sought to implement elements of that strategy via the land use
planning system.

The vision of MSW is that of making space for flood water rather than defending against it. Many coastal and inland areas
have been regularly inundated, but defending against all flooding in these areas had become unaffordable. The MSW strategy
is integrated with related regionally applicable policies including the "*Coastal Erosion Risk Management Evidence Plan*",
"*Catchment Flood Management Plans*" and "*Directing the Flow: Priorities for Further Water Policy*" (Evans et al., 2004;
Thorne, 2014) that encourage practitioners to deliver more sustainable flood management, which also considers water quality,
biodiversity and engagement with rural communities. PPS25 is a land use planning policy, applicable at a site-specific level
that provides guidance on how planners and developers should address flood risk. It includes a risk-based sequential test
intended to direct development to areas of lower flood risk. PPS25, and the 2014 "*National Planning Policy Framework*" that
superseded it, adopt a broader Sustainability Appraisal (SA) of flood management, considering economic, social and
environmental impacts. They require consideration of the spatial distribution of flood risk and how that risk distribution
changes in response to proposed mitigation measures, so as maximise net benefit of flood management.

In 2010, "*Surface Water Management Plans*" (SWMP) were required by government under the "*Flood and Water Act*". The
SWMP outlines a preferred surface water management strategy indicating how flooding from sewers, groundwater, and non-
main rivers will be managed. The adoption of sustainable urban drainage systems (SuDs) is encouraged to deal with runoff
following intensive rainfall (Defra, 2010). The SWMP works alongside PPS25, assessing flood risk to inform local authority
planning decisions, which are now required to ensure that new development does not increase flood risk. Flood risk modelling
and mapping are generally conducted by the environmental regulator, the Environment Agency (Environment Agency, 2014c),
on behalf of local authorities who have been made the Lead Local Flood Authority (following the Pitt (2008) review which
identified institutional complexity as a major barrier to addressing flooding in the UK). The maps of flood probability are
used in flood risk assessment and input to strategic planning. These policies and practices also encourage the participation of
public and NGOs stakeholders in the development of SWMPs.

The subsequently developed "*National Planning Policy Framework*" (NPPF) (DCLG, 2012) aims to restrict inappropriate
development in areas at risk of flooding, directing development away from areas at highest risk, but where development is



necessary, making it safe without increasing flood risk elsewhere. This requires a strategic flood risk assessment, an assessment by one or more local planning authorities to appraise the current and future flood risk from all sources (surface and ground
waters, fluvial and coastal) and with consideration of possible impacts from climate change. The NPPF represents an extended version of PPS25 intended to more comprehensively assess the impact that land use change and development will have on flood risk. For example, local planning authorities use flood risk information (i.e. Flood map for Planning – Rivers and Seas) provided by the EA (Environment Agency, 2014a), to consider opportunities for reducing flood risk to both existing communities and new developments. The NPPF also ensures that emergency planning capability is evaluated against the
forecast flood risk (DCLG, 2012). The EA also provide a live flood warning map (indicating flood alert, warning, and severe warning) showing locations at risk (Environment Agency, 2014b). Through this public release of flood risk information, the intention is that public awareness, preparedness and participation will be enhanced. Current UK flood policy thus seeks to integrate FRM with land-use planning, considering future development and flood risk, which addresses social, economic and environmental criteria. Whilst this may be an example of good practice in strategic and sustainable FRM, the UK still has room
to improve in aspects such as cost and time effectiveness, and in its complex governance structure (Green, 2014).

**2.3 The Netherlands**

The total land area of the Netherlands (NL) is about 34,000 km$^2$ of which more than 67% is situated below mean sea level (Beck, 2012). The safety standard of dike-rings and other measures is legislated for and reviewed every five years. The population is about 16.8 million with over 8.9 million properties located on flood prone areas in 2012 (Jongman et al., 2014).
As a result, land-use is intense, limiting flood management options, and placing great reliance on engineered coastal flood protection measures (Wesselink et al., 2007). It is vulnerable to coastal flooding and large parts of the country are still subsiding (van Stokkom et al., 2005). Spring ice melt from the mountainous region of the upper Rhine and Meuse rivers exposes the country to fluvial and flash flooding, with major floods in 1993 and 1995 (Vis et al., 2003; Wind et al., 1999). The country has a millennial history of flood management including from the 14th century the building of dyke-rings around polders to protect
land and settlements from flooding. Today there are more than a thousand polders protecting 65% of the Dutch coasts (Van Stokkom and Witter, 2008).

The 1953 North Sea Flood, which caused 1,800 deaths, spurred the NL to develop a high coastal flood safety protection standard. The country has the highest flood protection standards in the world, with 1-in-4,000 to 1-in-10,000-year flood return period infrastructure (i.e. coastal dike-rings), protecting populations and economic activities, especially in the West Coast
cities of Rotterdam, the Hague and Amsterdam (Klijn et al., 2004; Gerritsen, 2005). Despite the efforts of the Dutch government, the 1993 Meuse and Rhine River flood required over 10,000 people to be evacuated, with damage costs in excess of 10 billion Euros. The Dutch government understood it could not rely wholly on engineered flood defences to achieve an acceptable degree of risk (Olivier and Wytze, 2006; Wind et al., 1999) and in 1999 introduced a new water management policy, "*Room for Rivers* (*Ruimte voorde Rivier*)" (Böhm et al., 2004; Van Stokkom et al., 2005) (see Table 1). This required major



changes in FRM, including that: (i) water had to be guided in the landscape following an explicit spatial planning process, and

(ii) water had to be retained, stored and when necessary, land drained.

**Table 1: Features and functions of the "*Room for Rivers*" policy in the NL**

| Features | Functions |
|---|---|
| (i) Awareness | The Dutch government needs to improve communication on the nature and scope of risks and, in addition to its own efforts, to offer all citizens the opportunity to contribute or participate in risk reduction e.g. floods preparedness. |
| (ii) Three-steps-strategy | The need for a robust and resilient approach to ensure safety and reduce water-related problems, based on the following principles:<br>• Anticipating instead of responding.<br>• Not passing water management problems on to others, by following a three-step strategy (retention, storage and discharge).<br>• Allocating more space to water in addition to implementing technological measures. |
| (iii) Giving rooms to rivers | Encourage water storage. |
| (iv) Spatial planning | Adapting the spatial zoning strategy to prevent any human activities in the floodplains to interrupt the river discharge capacity |
| (v) Knowledge exchange | Encourage social learning and public education relating to water and river management. |
| (vi) Governing responsibilities | The provincial and local municipal authorities and water boards all need to share their responsibilities and ensure water related safety problems i.e. flood risk. All institutions need to ensure the effectiveness in FRM. |
| (vii) Investments | It requires additional fund and investment in FRM systems for the projected climate change and land subsidence. |
| (viii) International or transboundary co-operation | Co-operation with other shared-river basin countries (i.e. Deutschland and Luxemburg) on FRM should be intensified. |

Sources: Adapted from Klijn et al., 2008 and Van Stokkom et al., 2005

Along with the long history of flood management, Dutch water boards that are responsible for FRM have an established

engineering tradition which, it is argued biases them to management options with which they are familiar (Klijn et al., 2008).

Some critics thus argue that the "*Room for Rivers*" policy remains over reliant on dike-rings, levees construction and other

river regulation engineering (see Figure 1) (Hudson et al., 2008; Ten Brinke and Bannink, 2004). Nevertheless, the "*Room for*

*Rivers*" approach does represent a major shift in FRM, with strategies aimed at integration of floodplain development and

spatial land-use planning to meet socio-economic needs, ecological conservation and awareness of biodiversity, and wide

stakeholder involvement. Collectively this novel approach is considered to represent a progression from FRM to sustainable

FRM (Van der Brugge et al., 2005; Van Stokkom and Witter, 2008).




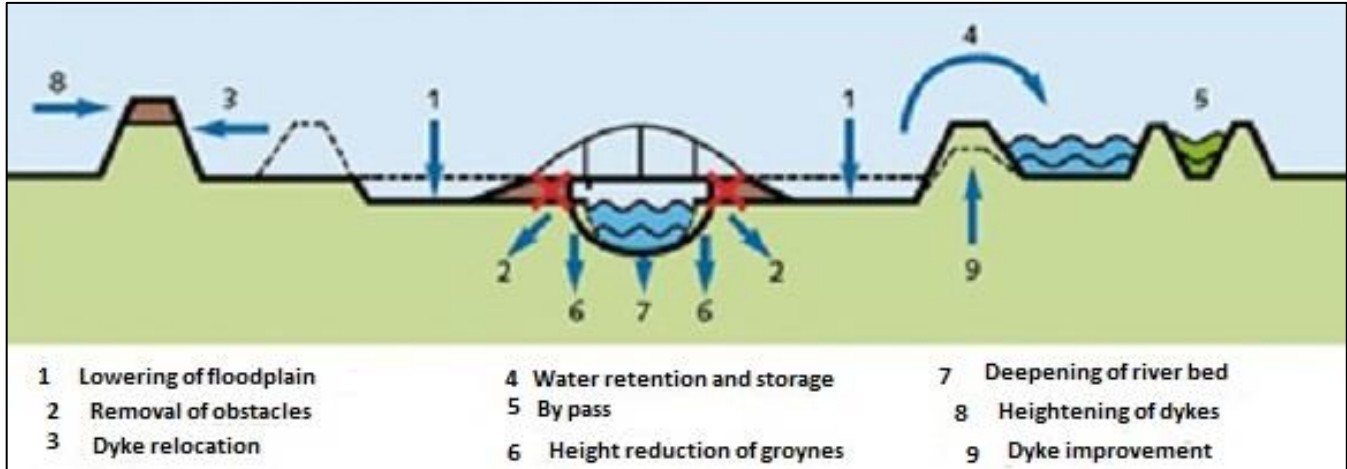

| 1 | Lowering of floodplain | 4 | Water retention and storage | 7 | Deepening of river bed |
| 2 | Removal of obstacles | 5 | By pass | 8 | Heightening of dykes |
| 3 | Dyke relocation | 6 | Height reduction of groynes | 9 | Dyke improvement |

**Figure 1: Measures of "*Room for Rivers*" in the NL. Source: Room for Rivers Program office.**

## 2.4 The United States

The United States (US) regularly experiences severe flooding. According to the US National Weather Service (NWS), there has been an average of 87 deaths per year from flooding between 1989 and 2018 (National Weather Service, 2018) and 29 flood events during the same period that resulted in greater than $1 billion in economic losses, with an average event cost of $4.3 billion. This does not include flooding caused by tropical cyclones (42 during the same period with an average event cost of $21.9 billion). (National Atmospheric and Oceanic Administration, 2019). Among the most disastrous events in this time

period are Hurricanes Katrina, in 2005, Sandy in 2012, Harvey, Irma, and Maria in 2018, and Florence in 2019, along with flooding in the Midwest US in 1993, 2008, and 2019 (Chan, 2004; Link, 2010; Xiao et al., 2011; Vance, 2012).

In the US, large populations are located along the banks of watercourses, lakes and coasts. In coastal flood prone areas of Florida, California, Texas, Louisiana and New Jersey, populations have expanded greatly since the 1980s (Niedoroda et al., 2010); the US Census projects an additional 82 million residents by 2030, an increase of 29% of the current population in

coastal areas such as Florida, Virginia, New York States in the East Coast and Mississippi in the Mid part and California in the West Coast (Hamin et al., 2019; Maantay and Maroko, 2009). Demand for development land remains high, and along with climate change, is likely to increase flood risk, especially along coastal areas (Bagstad et al., 2007; Aerts et al., 2013; Burgess et al., 2007).

The federal government has controlled the main flood management institutions since the 1920s when it took responsibility for

managing floods, primarily through flood control structures. However, at that time flood legislation was unclear, particularly with respect to the relationship between federal, state and local government. Notably, the federal government sought to share the financial burden of flooding and flood protection with state governments and local communities, and the percentage to be covered by state and local governments has increased over time. Evan as more funds were committed to flood control works, studies and events showed that flood losses were not declining which led to call for flood management rather than flood control

(Wright, 2000). Among the results of this is the National Flood Insurance Act signed into law in 1968, which had a carrot and





stick approach with the carrot being provision of flood insurance in communities that regulate development the 100-year floodplain, the stick. This was followed in 1977 by Executive Order 11988: Floodplain Management, issued by President Carter, which directed federal agencies to take the flood hazard into account when planning, funding, and implementing developments in flood prone areas (Arnell, 1984). Further, the Federal Emergency Management Agency was established in

1979 in order to coordinate under one agency all of the tasks associated with emergency management, including mitigation and the National Flood Insurance Program (NFIP).

Even with all of this, the federal management strategy maintained a focus on hard-engineering solutions. For example, the 1993 Mississippi River flood, which led to the river flooding over 1,200km of channel and about 840,000 $km^2$ in area (Figure 2), resulted in over \$15 billion in economic damages, provided an impetus to review the US approach to flooding. The

subsequent "*Galloway Report*" in 1994 proposed that development in the floodplain should be avoided unless no alternative locations existed (Galloway et al., 1999), similar to the "*MSW and Room for Rivers*" in the UK approaches described above. Yet, while the report was in preparation, reconstruction of damaged and breached levees was ongoing. However, the Galloway report indicated that whilst embankments and levees are important in protecting urban areas, they are insufficient on their own. Floodplains should be managed as part of the natural ecosystem, with risk-based forecasting used to inform flood management.

These practices encouraged sharing of flood risk information with the public and improved awareness of and preparedness against flood risk. From the mid 1990's there were improvements in practices of: (i) hydrological data collection during and after floods; (ii) development and installation of better instruments to evaluate coastal and river flood risk (e.g. Use of GIS, remote sensing and GPS to identify and understand flood hazards in various locations); and (iii) development of hydrologic models for more precise flood monitoring and prediction (Changnon, 1998). Since then, among other publicly available risk

information sources, the NWS provides river level data with maps showing potentially flooded areas through its Advanced Hydrologic Prediction System (AHPS; https://water.weather.gov/ahps/) and the United States Geological Survey (USGS) monitors river levels and discharges for thousands of rivers nationwide (water.usgs.gov).


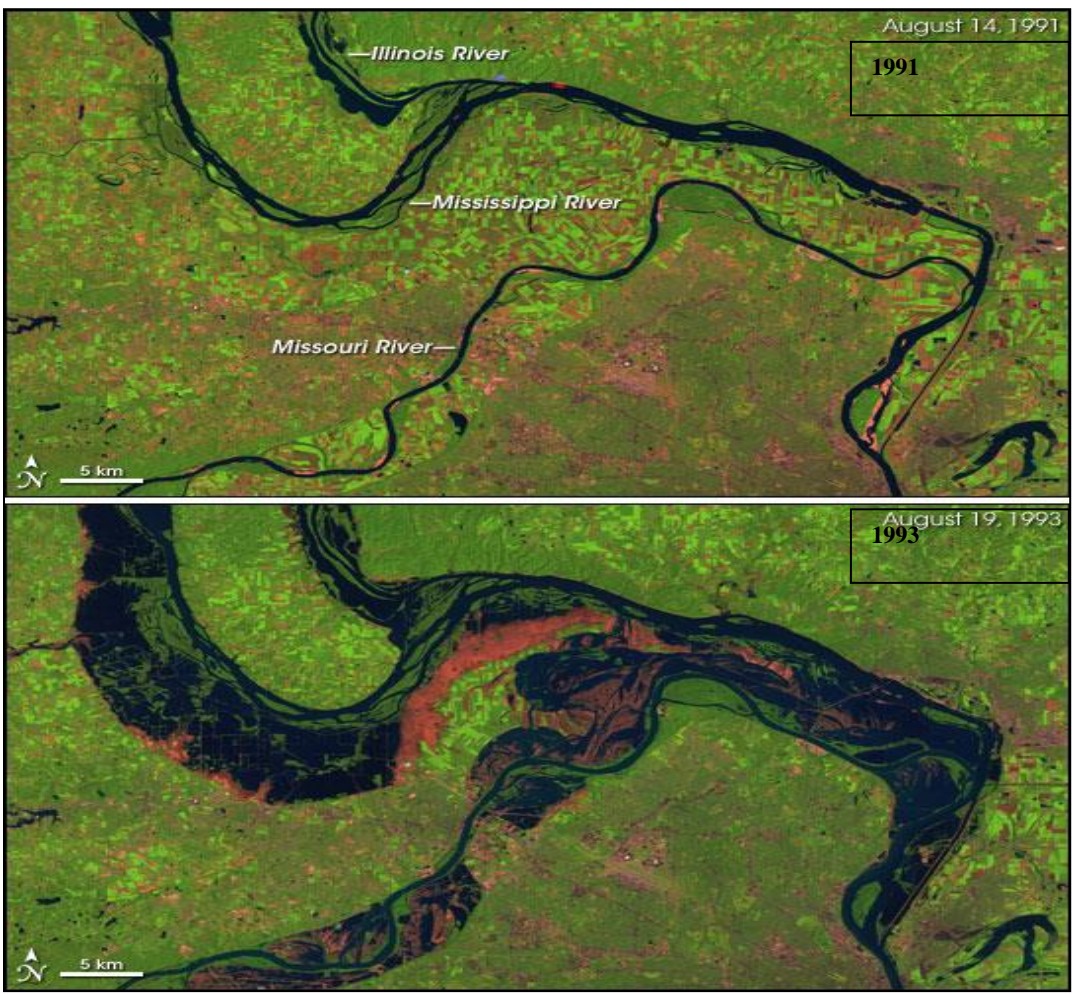

**Figure 2: Lower image of 1993 flood condition compares to the normal condition in 1991 upper image in the lower Missouri, the**
**Mississippi River. Source: (Allen J, NASA).**

The NFIP marked a particularly significant development of flood management in the US. The scheme, the world's largest

national flood insurance program, provides short- and long-term financial assistance to residents in flood zones (Arnell, 1984).

The program enables property owners in to purchase insurance protection, administered by the Federal government, against

losses from flooding, and requires flood insurance on all properties in the designated 100-year floodplain as shown on "*Flood*

*Insurance Risk Maps*" (FIRMs). Despite the requirement for flood insurance and the potential sanctions for not having it if

flooded (ineligibility for other federal disaster assistance), approximately only 50% of properties in the designated 100-year

floodplain carry flood insurance (National Association of Insurance Commissioners, 2017). This may reflect several factors,

including a misunderstanding of property owners of their flood risk and an expectation that disaster relief will be forthcoming

despite the requirement.



The NFIP aims to provide flood protection for property owners and discourage development in substantial risk areas by limiting

access to insurance. The NFIP is sponsored by the Federal government which also provides insured residents also have access

to emergency financial relief aid should they suffer flood damage (Longenecker, 2008). The program was designed to be

financially self-supporting, but the US Government Accountability Office reports that losses cost the taxpayers about USD

$200 million annually, and that since 1978, the NFIP has paid more than USD$51 billion in flood claims. The Congress

originally intended that the NFIP program be supported by premiums, but it is not for various reasons including subsidised

insurance rates for pre-existing structures and repetitive loss claims for many structures with no action to reduce their risk.

Further, sufficient funding from Congress for FIRM updates has not been forthcoming despite the fact that many of the maps

were created in the 1980s and thus do not reflect floodplain changes over time, whether from upstream development or climate

change.  Insurance is one means to mitigate flood risk but designing and delivering successful schemes are evidently difficult,

with issues arising relating not just to affordability, but cost sharing, sacrifice of very high-risk areas, and insurance industry

returns and expertise (Ball et al., 2013; Crichton, 2008; Michel-Kerjan and Kunreuther, 2011).

As is the case in many places, attention to flood risk increased with an event that forces communities to recognize the nature

of the risk it faces. For example, following Hurricanes Katrina and Sandy, US authorities responded with flood sensitive

strategic plans.  In the aftermath of Sandy, New York City adopted a "*rebuild by design*" coastal master plan that integrates

climate change into an adaptive development planning process (Rosenzweig and Solecki, 2014). This includes production of

flood risk maps to increase public awareness; development of an emergency contingency plan for all city districts with specific

attention on vulnerable social groups (e.g. minorities and the elderly); and raising flood protection for particularly vulnerable

(Aerts et al., 2013). These strategic actions, together with those described above, indicate the US has adopted a "mixed" options

approach to dealing with rising flood risk.

However, the devastating Houston flood following 2017's Hurricane Harvey highlights that not all US cities are adequately

prepared. Houston has exacerbated its flood risk through fast, sprawling development that has led to substantial loss of wetland

storage, expansion of impermeable surfaces, and lack of related investment in flood defence infrastructure. This is reflective

of the need for better planning and co-ordination between federal, state and local governments. While flood management in

the US has historically been a federal responsibility, the NFIP is only successful as a shared responsibility. However, with this

change, insufficient attention has been given to who bears the costs of flooding. In Houston's case, the residents and federal

government have borne the costs while the benefits of development have gone to the city's tax base and to private developers

(Berke, 2017). Thus, while the US has made important and successful strides toward SFRM, more is needed to achieve its

goals.

## 2.5 Japan

Japan covers approximately 378,000 km2 of which 70% is hilly or mountainous land, hence low-lying flood prone areas have

been preferentially developed for settlement. In this decade during 2010s, across the country as there are more than 49% of

the population, about 60 million people, reside on floodplains (Huang, 2014). The country unfortunately has got frequent flood





hazards and disasters records as the topographic feature of Japan tends to have short steep rivers with little upstream storage. Hence, flash and pluvial and combined type of floods are particularly occurred often. Over the last 30 years intensive rainfall

events (>50mm/hr) have increased in frequency by about 50%, and those >100mm/hr have more than doubled (Yamada et al., 2011) and it is expected that this trend will continue with climate change. Japan has flood hazards arising from typhoons, torrential rains, snowmelt and tsunami, and past floods have been associated with major impacts (Fujita and Hamaguchi, 2012). For example, in 2000 at Tokai city in Nagoya, pluvial flooding killed 18 people and injured 115, and economic losses were about 978 billion JYP (c. $9.57 billion USD). Another example, in 2004 during the Niigata-Fukushima flood, a result of

torrential rain, killed 20 people, and inundated 26,000 properties, making 5,800 homeless. In the same year, four strong typhoons (Songda, Meari, Ma-On and Tokage) hit the East coast of Japan between September and October causing sea surges, with 180 killed or missing and over 23,000 properties destroyed (Zhai et al., 2006).

Historically, flood management policy was haphazard in Japan, and it was not until the 1961 "*Disaster Countermeasures Basic Act*" that legislation provided the basis for an integrated disaster management strategy and clearly defined responsibilities

across the government. Specific laws relevant to flood prevention, such as the "*River Law*" followed in 1964 which reorganised river administration and improved flood governance. The Water Law was the catalyst for the 1977 "*Comprehensive Flood Control Measures*" policy that specifically focused on flood prevention, flood control and response to flood events (Takahasi, 2009). Under this policy, rivers are divided into three classes (A-C); class A rivers, the largest in terms of area, length, and significance of their assets (economic and population in their basins), are managed by the Ministry of Land, Infrastructure and

Transport (MLIT)'s River Bureau which reports directly to the Japanese Central Government. Flood risk in the smaller Class B and C rivers is managed at municipal and local government levels, with MLIT support (MLIT, 2008). Ikeda et al. (2008) note that whilst flood fatalities fell after 1960 as new flood protection policies took effect (including an MLIT policy that 1% of national income be invested in flood measures from 1960–1990), economic losses from flooding have not fallen and remain high.

Kundzewicz and Takeuchi (1999) illustrated since the River Law was enacted, the MLIT and related institutions employed a hard engineering flood protection approach, with the main flood control strategy being to transport water quickly to the sea. Super levees, divergent canals, flood-ways and bypasses were constructed. This approach was questioned after the 1977 Nagasaki flood where 375 people in the unprotected upstream area died following a 180mm/hr rainfall event. Two problems were particularly evident: (i) flood control or hard flood protection measures did not cover all parts of the river, due to the cost

of such defences; and (ii) class A rivers and class B/C rivers were managed by different institutions with inadequate communication between them and a lack of integrated flood risk appraisal (Ueno, 2002). Afterwards, in the 1990s, the River Law recognised the complex nature of integrated catchment management, with the law seeking to address objectives related to flood risk, water resources and environmental quality and legislated in 1997.

"*Article 1. The purpose of this Law is to contribute to land conservation and the development of the country, and thereby*

*maintain public security and promote public welfare, by administering rivers comprehensively to prevent occurrence of*



*damage due to floods, high tides, etc., utilize rivers properly, and maintain the functions of the river water by conserving the fluvial environment*".

This amended River Law reflected that the government was advocating a more sustainable approach to flood risk, with integration of social and environmental issues into FRM, as well as wider public engagement (MLIT, 2008):

"*When river administrators intend to draft a river improvement plan, they shall consider opinions from persons with experience or an academic background when necessary*" (Article 16-2-3), and "*In connection with the previous paragraph, river administrators shall take necessary measures, such as public hearings etc., to reflect the opinion of the people concerned whenever necessary.*" (Article 16-2-4).

In 2000s, the MLIT issued their "*Effective Flood Management including Basin Responses*" policy, which emphasised flooding
is unavoidable and accepted the nature of flooding in water-prone areas (e.g. wetland) to enhance ecological value (Ikeda et al., 2008). This policy not only focused on the areas with high assets (e.g. urbanised floodplain), but extended to cover the rural areas in the river basin (i.e. upstream) (MLIT, 2008). Key to this policy is the integration of hard and soft flood protection measures, rather than reliance on traditional engineered defences. Whilst, Japan understood that the importance of flood protection especially to cities owing to flood consequences by substantial populations and affiliated economic assets. This
effective flood management policy also coping with the "*Act on Countermeasures against Flood Damage of Specific Rivers Running across Cities*" (legislated in 2003) that particularly targeted for reducing flood risk for various sized (large to small) rivers that running across the cities and towns (MILT, 2003). The Act has further amended in 2004 and further legislated the "*Flood Fighting Act*". These two amended Acts allocate the municipal governments understood their roles explicitly on delivering according FRM practice (Yamada et al., 2011). For example, informing residents for gaining awareness by issuing
"*flood warning*" for the communities understand potential flood spots. Whilst, also providing emergency services (e.g. temporary shelters, evacuation medical services, etc.) for enhancing emergency response and recovery practices (OECD, 2006). Lately towards 2010s, the Japanese authorities further addressing flood resilience approaches and especially tackling the perception via issuing flood hazard maps (MILT, 2008). The MILT further concerned about the intensive rainstorms enhanced urban floods and legislated the specific measures to reduce flood risk during heavy rainstorms (MILT, 2013). After
the Tsunami and coastal floods in early 2010s, the MILT has further implemented the "*Act on Special Measures concerning Urban Reconstruction*" in 2018 to strengthen the flood resilience, particularly tackling the prevention, emergency response and recovery processes (see Figure 3). This is used to support development decision making by land use planners by setting up the land use regulation zone system, direct the flood proofing of existing urban facilities (e.g. the underground, railways, public services facilities), raise flood awareness amongst the at-risk public (via flood warning system, hazard map and future
flood projection), and inform the emergency response and evacuation procedures (via relocation practice) of the civil emergency services (MILT, 2018).



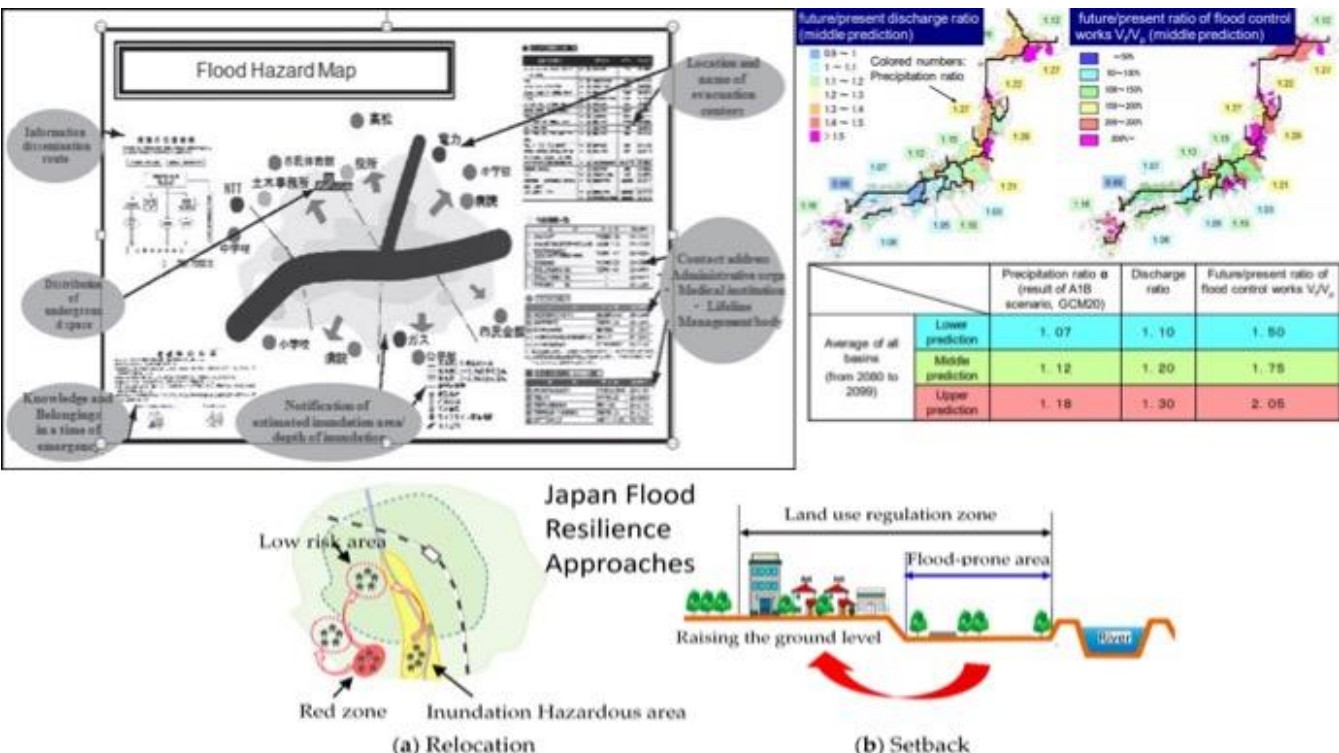

**Figure 3: Flood resilience approach in Japan – Upper: Flood Hazard Map and Fluvial Flood Prediction with future Climate Projection; Lower: Japan Flood Resilience approach for a. relocation and b. setback strategies to protect residents (adopted from MILT 2008;2014; Fan & Huang, 2020).**

## 3 Discussion

### 3.1 Sustainable flood risk management (SFRM) – where are we now?

"*Sustainable Flood Risk Management* (SFRM)" evolved during the 1990s, at the time when the concept of Sustainable Development (SD) became prominent (Brundtland, 1987). The three pillars of sustainability (social, economic, environment) are widely recognised and influential to policies (UNCED, 1992). SD strategies seek to ensure economic development is conducted in a manner that respects environmental limits and values, and considers the distribution of all costs and benefits of development through time (inter-generational equity) and across social groups (intra-generational equity) (Pearce et al., 1996; Sneddon and Fox, 2006; Morse, 2008). An additional imperative is the development of strategies through open and participatory mechanisms. These sustainability principles are applicable to all types of development, considering for the inclusion in development of FRM strategies, so called Sustainable FRM (SFRM) (Evans et al. 2004; Hooijer et al., 2004).

Table 2 illustrates how SFRM is interpreted differently across several European nations, and reveals that whilst SFRM is not a contested concept, it does lack a widely accepted definition. Similarly, De Bruijn et al. (2007) suggest that SFRM could be understood simply as the ability of society and ecosystems to cope with several types of flood risk whilst maintaining the level





of well-being, whilst Chan et al., (2013b) attempt to characterise SFRM practice within a more explicit sustainability

framework. Nevertheless, for the four countries we discussed above, progress from flood control, to FRM, and now towards SFRM is clearly evident (see Figure 4).

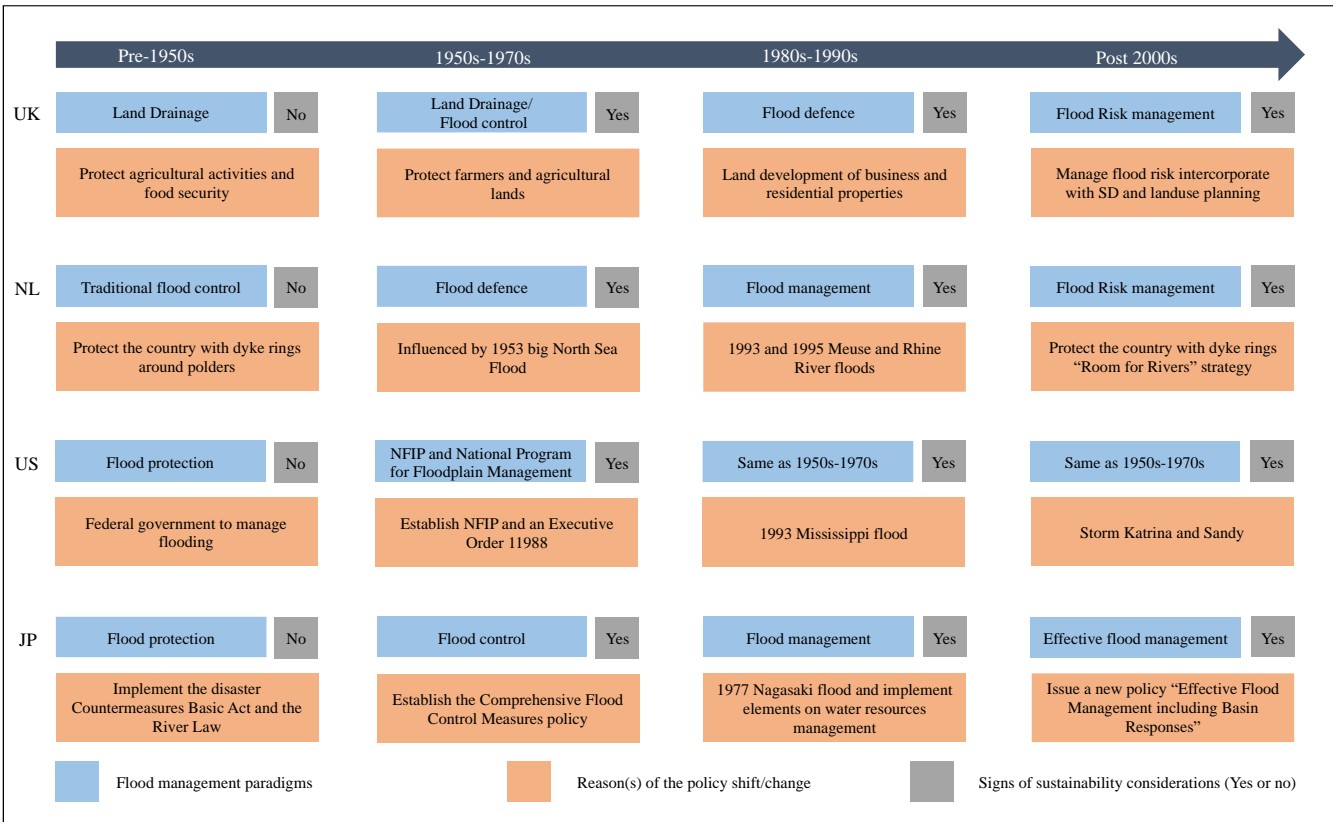

**Figure 4: Flood management paradigms and major changes towards SFRM.**

Examples of action taken towards SFRM include the open provision of flood risk information to aid participatory planning

(post 2000 in NL), flood mitigation measures subject to wider economic appraisal such as extended CBA considering environmental impacts (UK), cost sharing through national flood insurance schemes, and design of flood mitigation with nature (1950s – 1970s in US), both to protect the natural environment, and to recognise its value in mitigating flood risk (e.g. via sustainable drainage systems, wetlands, reintroduction of river meanders) (Butler and Pidgeon, 2011; Green, 2014; Porter and Demeritt, 2012; Scott et al., 2013). From our review, it is apparent that no country has FRM that comprehensively addresses

sustainability concerns (see Table 3).

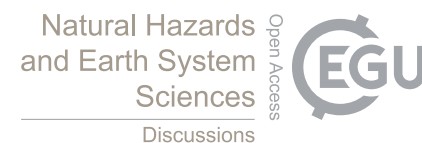
**Table 2: Definitions and principles of SFRM in different countries**

| Countries | Legislative documents | Definition of SFRM |
|---|---|---|
| UK: Scotland | FRM (Scotland) Act 2009 (the FRM Act) | "Sustainable flood management provides the maximum possible social and economic resilience* against flooding**, by protecting and working with the environment, in a way which is fair and affordable both now and in the future." |
| UK: England and Wales | Flood and Water Act (2010)<br><br>Defra (2011) | "In exercising a flood or coastal erosion risk management function, an authority must aim to make a contribution towards the achievement of sustainable development."<br><br>Sustainable development in the context of flood and coastal erosion risk management (FCERM) includes:<br>• taking account of the safety and wellbeing of people and the ecosystems upon which they depend,<br>• using finite resources efficiently and minimising waste,<br>• taking action to avoid exposing current and future generations to increasing risk, and<br>• improving the resilience of communities, the economy and the natural, historic, built and social environment to current and future risks. |
| European Union | EU Floods Directive (2007) | The EU Floods Directive (2007) has aimed for:<br>(i) ensuring quality of life by reducing flood damages by being prepared for floods;<br>(ii) mitigating the impact of risk management measures on ecological systems at a variety of spatial and temporal scales;<br>(iii) the wise use of resources in providing, maintaining and operating infrastructure and risk management measures; and<br>(iv) maintaining appropriate economic activity (agricultural, industrial, commercial and residential) on the flood plain |

Note: (* 'resilience' means: 'ability to recover quickly and easily'. The Scottish Executive uses it to deliver the 'four as': Awareness + Avoidance + Alleviation + Assistance.)
(** flooding means all types of flooding: surface (pluvial), sewer, river, groundwater, estuarine and coastal)

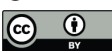

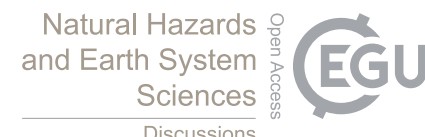

**Table 3: Strengthens and weaknesses of current FRM practices towards SFRM across 4 countries**

| Pillars | Strengthens of practices | Practices | Facts/examples | Weaknesses of practices | Arguments and reasons |
|---|---|---|---|---|---|
| Social-economic | Social justice and equity | **UK:** Pitt (2008) reported to address social justice and equity issues in FRM | Defra research (projects FD2605 and FD 2606) has looked at issues of inequity of FRM in England | Social injustice and inequity | **UK:** No exact implemented guideline to address issues of inequity of FRM, but most of poors live in the flood zone (Johnson et al., 2007) **US:** Ethical minorities - A research showed the minorities in NYC mostly are not insured by the NFIP flood insurance scheme (Maantay and Maroko, 2009) |
| | Flood risk information on social-economic impacts | **UK, NL:** Flood risk mapping (EU, 2007; Environment Agency, 2014abc); **US:** Interactive flood information map (NOAA, 2014ab); **JP:** Flood hazard map (MILT, 2008) | Increase understanding of flooding, preparedness and awareness; assess potential economic impacts (UK and NL); showing flood histories (hazards), types of flooding, current flood risk and level of protection (US); showing the post-flood emergency routes and contingency plan (JP) | Analysis and reporting of potential flood risk on possible social economic impacts | **UK:** Flood risk mapping – may be too technical for the public to understand (Porter and Demeritt 2012), absence of the detailed information (e.g. protection measures) on the flood maps |
| | Cost-benefit analysis (CBA) | **UK, NL:** Adopted CBA in NPPF and Room for Rivers (Carter et al., 2009; Eijgenraam,2005) | Maximising the cost and value, and create economic justification on FRM process | Cost-benefit analysis (CBA) | **UK, NL:** Difficult to evaluate environmental and social aspects in the CBA |
| | Flood insurance | **US:** NFIP (Arnell, 1984) | Most of residents (who live within 1-in-100 years flood zone) are covered in the NFIP scheme | Flood insurance | **US:** Poor people and minorities are mostly not insured by the NFIP scheme (who live outside the insurance coverage boundaries - the risk is higher than 1-in-100 years) For example, some residents were the poorest community in New Orleans lived in the Lower Ninth Ward, which was flooded by a catastrophic breach (during Storm Katrina) that without NFIP coverage) (Chamlee-Wright and Storr, 2009; Burby, 2006) |
| | Apply sustainable flood management practices | **UK:** Adopted Sustainable urban drainage systems (SuDs) (Mitchell, 2005; Coupe et al., 2013) | Reducing flood risk, surface water pollutant level, the pressure of the surface runoff discharge, improving amenity and environmental values (e.g. improve wildlife habitats) | | |
| Other issues | Governance | **UK:** Better FRM governing structure | | Complex institutional arrangement | **UK:** Time consuming and not cost effective (Green, 2014) |



For example, Maantay and Maroko (2009) show how in New York City, the national flood insurance scheme does not effectively reach some social groups, particularly ethnic minorities, who tend to be exposed to above average flood risk. Social equity issues tend also to be under represented in SFRM studies, although researchers and practitioners are increasingly alert on how resilience to flooding varies spatial-temporally (Yang, et al., 2021) and socially with low income households a particular concern. Similarly, economic appraisal of flood strategies recognises environmental impacts, but ecosystem service values, and the wider benefits of nature-based flood management (Dadson et al. 2017) are nowhere routine in such economic appraisal. With respect to flood management, the Pitt review into a series of major floods in England (Pitt, 2008) highlighted that flood governance can be a major problem; so many organisations had responsibilities for FRM, and at a variety of geographic scales and flood types, that a high degree of institutional complexity resulted which acted as a barrier to effective FRM, as also seen in the US. Local authorities have since been given the lead role in FRM.  Porter and Demeritt (2012) commend the openness and transparency of flood risk mapping but raise concerns over the degree of technical expertise needed for the public to understand and act appropriately on the information conveyed. These examples indicate that in the four countries we reviewed, challenges to SFRM exist. However, these tend to be challenges of an operational rather than philosophical nature, challenging the delivery of SFRM but not the principle. That is, there is now a widespread recognition of the need to address sustainability concerns and embed sustainability principles into FRM policy and practice.

## 3.2 Implementing SFRM in Asian coastal megacities

Currently, many Asian coastal megacities are operating predominantly within the flood control and defence paradigms, including Guangzhou, Shenzhen and Hong Kong (Chan et al., 2013; Yang, et al., 2018), Shanghai (Balica et al., 2012), Bangkok (Keokhumcheng et al., 2012), Ho Chi Minh City (Storch and Downes, 2011; Nguyen, et al., 2021), Jakarta (Texier, 2008; Wannewitz and Garschagen, 2021) and Singapore (Chan et al., 2018; Chan et al., 2021). These cities all feature in the top coastal cities at risk by 2050s (Hallegatte et al., 2013), due to their growing populations, economies, and rising flood hazard. Limited room to expand has resulted in development on floodplains, wetlands, and reclaimed coastal areas (Chan et al., 2014; Ji, et al., 2021), a practice common in Asian coastal cities (Fuchs et al., 2011) and we discuss the progress of their flood management strategies in this section (see Table 4).

Looking at the past (before 1990s), Singapore has suffered from severe flood due to urbanisation since the nationhood in 1965 (Chan et al., 2018). The Drainage Department at Singapore was established in 1972 to prevent floods. Singapore government invested heavily to construct dense networks of drains and canals is the major approach for flood management before the 1990s (Lim, 1997), which was effectively reduced flood prone areas from 6900 ha in 1960s to 207 ha in the 1990s (Lim and Lu, 2016). Owing to economic development, Singapore was pioneered adopted Low Impact Developments (LIDs) approach (similar to SuDs in the UK) in the 1990s that is the evident the flood management practice has been transformed as discussed, for example constructed stormwater retention ponds for stormwater storage and reuse at Kallang district (Lim and Lu, 2016). Indeed, the neighbourhood coastal cities such as Bangkok, Jakarta, Ho Chi Minh City are also facing the similar issues (on urban floods) afterwards, particularly towards their fast urbanisation and developmental period during early 1980s. Rapid



landuse changes transform green spaces (e.g. agricultural and farmland, forest, wetland, etc.) into urban area, but land drainage and flood measures were unable coping with the urban runoff (Takeuchi, 1993; Jha et al., 2012; Huu, 2011; Katzachner et al., 2016). Mostly, the flood management practices before 1990s in these cities were mainly driven by flood control and defense measures. Jakarta was jurisdicted under Dutch influence on flood engineering foci in early 1920s, which demonstrated by the East Flood Canal Project was constructed during the Dutch Colonial period at 1924 and that was an extension of the Western

Floodway at the city of Jakarta to alleviate urban peak discharge (Jha et al., 2012). Bangkok was dominated by the engineering approaches such as engineering works for agricultural irrigation, embankments, reservoirs and drainage systems in 1980s and 1990s (Bouriboun, 1998). Across Mekong, Ho Chi Min City has also dominated by agricultural engineering works for the agricultural production (e.g. rice and poultry, etc.) and crops protection during the storms (e.g. typhoons) before 1990s (Huu, 2011). Likewise, Chinese coastal cities similarly faced urban floods that enhanced by urbanisation and fast developments

during the "*Open Door Policy*" established in late 1970s, such as urban floods in Shanghai during 1981 (Dou, 1991). Guangzhou experienced frequent urban floods due to condensed population density in the major district areas e.g. Tianhe, Baiyun, etc. (Zou, 2012). Engineering approaches (e.g. flood walls, dykes, drainage canals, pumping stations and dredging engineering works) were popular and applicable before 1990s in both cities (Zou, 2012; Meng and Dubrwoski, 2016).

During the 2000s, these coastal cities also gradually transformed considering wider aspects on social-economic risk and health

issues of the communities. For example, in Jakarta, the municipal authorities initiated non-structural measures including an early warning system, health service capacity building, and contingency planning including relocation and compensation schemes after the 2002 and 2007 floods as realised engineering works was insufficient to protect the communities (WHO, 2007; World Bank, 2009). In Bangkok, the Thai government still favourited using engineering works to further strengthen the flood engineering works (e.g. dykes, drainage system, canals and retention area) that focusing on improving the engineering

technology of flood mitigation, but also established the emergency response measures such as identified evacuation areas in Bangkok Metropolitan districts (Chen, 2007; Phamornpol, 2011). In Ho Chi Minh City, social economy reforms started in late 1980s and enhanced rapid urban expansion and deforestation that was with limited flood consideration, still rely on engineered measures (Krystian and Nguyen, 2005; Labbé, 2010), and later in the 1990s started considering flood relocation and relocated over 1 million people away from frequent flood zone (Danh and Mushtaq, 2011). In Shanghai, flood risk has recognised a

step-ahead as the municipality government understood the importance of learning flood risk analyses and information to evaluate the hazards and potential responses on measures because of frequent typhoons and relevant disasters (e.g. 2 to 3 times per year) (Lu, 2010). For example, Shanghai authority to raise the flood protection level of coastal defence from 1-in-100 to 1-in-1000 years protection level during early 2000s (Yin et al., 2015; Zhou et al., 2016). In Guangzhou, the municipality government similarly reacted after frequent floods in the late 1990s and adopted flood risk measures (Wong and Zhao, 2001).

The authorities then promoted the green infrastructure (e.g. via "*Green-blue network*" in Nan Sha District) to protect ecological and increased the hydrological risk understanding and recognised cultural value by conservation of heritage out of flood impacts (Timmeren, 2014; Han et al., 2015).



Thus, lately in the post-2000s, these cities have experienced more floods that urged the governments to progress and consider more wider with achieving social-economic and environmental pillars in the flood management, also have been influenced
from the global FRM practices. For example, Jarkarta established a Comprehensive Flood Management Plan that recognized higher risk from the consequences (e.g. populations, flood locations, etc.) (JICA, 2013). The authority adopted land use planning measures to avoid development elevating flood risk, and implemented a micro-insurance schemes and the relocation contingency plans to improve resilience (Jha et al., 2012). Similarly, in Bangkok, the Thai Government established the flood resilience strategy based on catchment management after the severe 2011 flood, which was the worst flood since 1942 with
46.5 billion USD damages and 680 deaths (Poaponsakorn, 2015) and recorded insurance losses solely from Japanese factories (e.g. Toyota, Honda and Nissan, etc. parts factories for engines) were 10 to 15 billion dollars and enhanced global insurers and governments understood about the chain effect of large floods (Meehan, 2012). The government initiated flood risk zoning policy and restricted developments in high risk areas (Water Resource Management Strategic Committee, 2012; Berkowitz, 2013; Supachai, 2016). In Vietnam, the authority has also established integrated flood management strategies (Eckert and
Huynh, 2016), but the municipality government still focusing on engineered flood measures (Katzschner et al, 2016). In China, Shanghai and Guangzhou governments have moved steps forward that aligned with the National Climate Change strategy. For example, Guangzhou has followed the National 12th Five Year Plan included a National Adaptation Strategy (NAS) for climate change (UNDP China, 2012) and established scientific warning system that based on accurate flood risk information (Lyu et al., 2016). Shanghai identified as one of the most vulnerable Chinese cities under climate change (Hallegatte et al., 2013;
Francesch-Huidobro et al., 2017; Yuan et al., 2017). The authority further established the flood monitoring and forecast systems were established and Meteorological Office worked with the IPCC on climate (e.g. sea-level rise) projections to further improve public emergency warning, planning and public preparedness (Li, 2015). Singapore is transforming from "*City in a Garden*" to "*City of Gardens and Water*". Singapore government also taking a role to further implement the "*Active Beautiful Clean*" (ABC) Waters Program for delivering sustainable and climate resilient measures on urban stormwater management
and adopted the "*source-pathway-receptor*" (SPR) model to address flood risk and climate change (Chan et al., 2018; Liao 2019).

In fact, Climate change is raising sea-levels making storm surges more hazardous (Nicholls, 2011), and is increasing the frequency, intensity and magnitude of storms (typhoons), intense rainfall events and sea surges (Webster, 2008; Webster et al., 2005). Natural resource extraction is also increasing flood hazard through the land subsidence it causes; for example, in
the coastal area of Bangkok groundwater extraction has resulted in subsidence of two metres since 1970 (Syvitski et al., 2009). The cities that are selected in this review rely upon hard engineered defences, but these structures offer a relatively low degree of protection. For example, the major urban drainage systems in Singapore have been improved from 1-in-50 years up to 1-in-100 years, whilst Guangzhou and Shenzhen only have a 1-in-20-year return period protection against typhoon and sea surge (Chan et al., 2012; Chan et al., 2018). For example, fatalities occurred in Hong Kong in 2010 when pluvial flooding
overwhelmed the 1-in-50-year level protection defences, whilst Shenzhen only has a 1-in-20-year return period protection against typhoon and sea surge (Chan et al., 2012). Such levels of protection are modest at best, and likely merit investment to





raise flood protection standards but even financially heavily invested and not guarantee flood risk could be substantially reduced, which is sensible the Asian coastal cities moved towards SFRM by learning from the global lessons and experiences and that destined ways forward on flood management extensively in other Asian cities.

**Table 4: Development of flood management practices in selected E. and S.E. Asian cities**

| City | Flood management practice in Asian cities relative to dominant flood paradigm in the West | | |
| | Before 1990s Flood protection (control and defense) | 2000s Flood risk management (FRM) | Post-2000s Sustainable flood risk management (SFRM) |
| --- | --- | --- | --- |
| Jakarta, Indonesia | Solely engineering response with Dutch colonisation influence (Jha et al., 2012). | FRM practice develops after 2002 and 2007 floods. Non-structural measures were implemented in the Urgent Flood Mitigation Project (WHO, 2007; World Bank, 2009). | Improving resilience e.g. established the Coastal Defense Strategy (JCDS) in 2014 (Hidayatno et al., 2017). |
| Bangkok, Thailand | Practice focused on engineering solutions (Takeuchi, 1993; Bouriboun, 1998). | Improved flood engineering technology (e.g. dykes, drainage system, etc.) and identified evacuation areas at the Metropolitan Administration (Chen, 2007; Phamornpol, 2011). | Established the community resilience plans, recognised flood risk after 2011 big floods (Berkowitz, 2013; Supachai, 2016). |
| Ho Chi Minh City, Vietnam | Irrigation canals and agricultural engineering works dominated to protect crops (Huu, 2011; Katzschner et al, 2016). | Doi Moi social market economy reforms and dominated engineered measures (Krystian and Nguyen, 2005; Labbé, 2010). In the late 1990s, initiated flood relocation scheme in flood zones (Danh and Mushtaq, 2011). | Integrated flood management established (Eckert and Huynh, 2016). |
| Guangzhou, China | Engineered measures to protect settlements (Zou, 2012; Meng and Dubrwoski, 2016). | Combat flood risk in various practices (Wong and Zhao, 2001); Promote ecological and cultural value (Timmeren, 2014; Han et al., 2015). | Guangzhou improved flood warning system and adopted Climate Change resilience plan (Lyu et al., 2016). |
| Shanghai, China | Flood management focused on engineering works (Dou, 1991; Ke, 2014). | The municipal government extended flood risk analysis and raised level of protection to 1 in 1000-year flood (Yin et al., 2015; Zhou et al., 2016). | Shanghai Meteorological Office worked with the IPCC on climate (e.g. sea-level rise) projections and improved resilience measures (Li, 2015). |
| Singapore | Singapore government invested on engineering works to alleviate floods in 1972 (Lim, 1997; Lim and Lu, 2016; Chan et al., 2018). | Pioneered in SE Asia to establish the LID practices after 1990s (Lim and Lu, 2016). | ABC Waters Program launched that based on BGI and LID in 2006 (Liao, 2019). Addressing climate change with SPR model (Chan et al., 2018 and Liao 2019). |

However, a key impetus of the shift in practice from the flood protection and defence paradigms, to SFRM, has been a recognition that the costs of traditional hard engineered flood defences are increasingly unaffordable, and that a wider package





of measures is needed to address flood risk. As learnt in the aftermath of the Hurricane Katrina disaster in New Orleans, FRM also requires better development of non-engineered measures, ranging from strategic land use planning with substantive public participation (Neville and Coats, 2009), specific attention to the most vulnerable communities (including insurance to aid recovery) (Chamlee-Wright and Storr, 2009; Burby, 2006), and well-prepared emergency and evacuation plans for when floods strike (Niedoroda et al. 2010). Experience with Hurricane Harvey in 2017 suggests that even if these lessons have been learnt, Flood risk may remain high due to a legacy effect of past land use planning and investment decisions.

Hard engineered flood protection measures will always be important in the defence of Asian cities, but whilst these defences can be more coherently planned (Francesch-Huidobro et al., 2017), evidence indicates that the level of protection needed is unaffordable (Jongman et al., 2014). Despite the apparent limitations of relying solely on structural defences, few efforts have yet been made to adopt a wider set of measures that incorporate non-structural protection measures, pre-emptive strategic and land use planning, risk awareness and communication, emergency planning, and post-event recovery and learning.

Current approaches also tend to focus on potential economic losses, neglecting the role and value of the natural environment and social considerations, such as impacts and recovery potential of different social groups, and participatory planning. Indeed, most Asian cities (even those with high flood risk) remain focussed on hard engineering solutions (refer to Table 4) and lack a sufficient range of climate change adaptation guidance and practice (Nguyen, et al., 2021), which may prove problematic as sea-level rises, and extreme storms, surges and typhoons become more frequent. Given the unaffordability of engineered defences necessary to mitigate their rising flood risk, Asian coastal megacities may find it advantageous to recognise wider international experience, and develop coping strategies that reflect a greater acceptance of options to '*Living with flood risk*' rather than assume that all such risk can be engineered away. Such coping strategies combine traditional engineering, soft engineering (e.g. SuDs), land-use planning, working with rather than against nature, and social strategies that recognise vulnerable communities, and engage stakeholders in co-production of responses to flood risk. The international experience clearly shows that SFRM approach is more complex than control or defend, hence significant operational and procedural challenges can be expected, with good governance needed to engage stakeholders effectively whilst avoiding undue institutional complexity.

## 4 Conclusion

In Asian coastal megacities, flood risk is high and rising, and defending against flooding effectively using traditional approaches is becoming financially unsustainable. Coping with flood risk, as illustrated by 'Room for river' type concepts, and through 'soft measures', as discussed above, is in some countries increasingly considered a necessary and more sustainable alternative to hard-engineered defences alone. The case studies in the four countries showed some good lessons to achieve long term SFRM direction to deliver flood management practices with social-economic and environmental concerns. We understand different countries and cities have their own interpretation on SFRM, but recommend policy makers to adopt





"mixed options" towards thinking about long term and sustainability that with social, economic and environmental
considerations.

However, this is a philosophy that has had relatively little influence on practice in the coastal megacities of Asia. There is of course, no prescriptive template for developing coping strategies, and each country and city will need and wish to develop measures appropriate to their specific contexts - physical, social and economic, environmental and cultural. For very dense coastal megacities 'making space for water' opportunities may be rather more limited than they are elsewhere in the world,
such that coping strategies will develop with different emphases. Once the limitations of hard-engineered defences are recognised and understood, sustainable development principals and tools can be used to shape coping strategies and help deliver more flood resilient cities.

**Author contributions**

FKS Chan is the lead author and contributes to writing, editing, designing, and reviewing the manuscript. Liang Yang
contributes to writing, editing, and reviewing the manuscript. Xiaohui Lu contribute editing and reviewing the manuscript. The other co-authors contribute to editing, graphic designs, and information collection.

**Competing interests**

The authors declare no competing interests.

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
