# Peer review of "Comparison of Sustainable Flood Risk Management by four countries: the United Kingdom, the Netherlands, the United States, & Japan, and the implications for Asian coastal megacities"

_Natural Hazards and Earth System Sciences, 2021_

## Referee Comment (RC1)

**Comparative analysis and implications of Sustainable Flood Risk Management in four front –end countries: the United Kingdom, The Netherland , the United Sates and Japan**

**(Anonymous Review Comments)**

**General comments**

This is a well informed and authoritative review paper which draws on and compares experience in 4 countries with respect to their transitions towards sustainable flood risk management, with the purpose of applying lessons learnt to cites in Asian coastal megacities. The paper is well structured and logically developed , cataloguing a lot of useful detail ( e/g table 2 an 3 and carefully reflecting on the change of emphasis s represented by each new measure, with a useful summary of flood management paradigms and change towards SFRM provided in Figure 4 . Tee title is somewhat misleading as it is unclear what is meant by "front end countries" and does not reflect the true intent of the paper ( in relation to SE Asia) . Overall it provides a historical overview across several jurisdictions , although perhaps more could be done to distil best practice from the 4 counties studied into consolidated practical guidance for SE Asia ( where contextual similarities allows this).

**Specific comments**

Title : Misleading and does not describe the full purpose of the paper. Unclear what is meant by "front-end cities" and reference to Asian coastal megacities should be made clear

Page 1: Abstract line 28-30: " This paper reviews the past and present flood management approaches and experiences from flood defence to FRM in four developed countries with the aim of highlighting lessons for developing mega deltas" . The paper does not fully explore the hydrological (and other contextual) dissimilarities between the regions being compared, and whether these dissimilarities can justify and sustain strategies from different parts of the world working elsewhere. Specifically rapidly "developing mega deltas" bring their own constraints which might be explored in more detail

Page 2 line 37 explain how subsidence arises from human factors ( e.g. as a result of excessive groundwater extraction)

Page 2-3 line 84 this focuses on SE Asian examples not reflected in the paper's title ( see earlier comment)

Page 3 line 86-87 reference is made to hard engineering solutions and flood control. This could be further explained both in respect to the engineering materials used ( e.g.concrete) and most importantly the basic driving principle of increasing in- channel conveyance .

Page 4: lines 111-112 " These experiences offer lessons from FRM in Asian coastal megacities". Whilst undoubtedly some of the reviewed changes in practice are relevant , such as managing urban flooding through the principle of source control (-no mention-), there are large hydrological ( and meteorological) dissimilarities between the areas being compared and this should be explored and acknowledged in more detail.

Page 4 line 120-130 Clearer distinctions could be made between rural and urban flood policy responses. Fore example the papers says little about the introduction of Sustainable URBAN drainage ( limited to line 153)

Page 5 line 135 " be more specific on what is meant by "externality effects"

Pages 16-17 – Tables 2 and 3 : information here represents a heavy UK focus

Page 5 : general comment: greater and more explicit distinction should be made between pluvial and fluvial flooding ( with respect to the strategies considered).

Page 6 line 180 " complex governance" structure ; fragmented responsibilities are a serious on-going in issues in UK flood management (e.g. see: Ashley R., Gersonius B., Horton B "Management flooding : from a problem to an opportunity" Royal Society Philosophical Transactions n A Volume 378 Issues 2169 April 2020

Section 2 The most recent references (around 2012-2014) seem somewhat dated with more recent papers on this topic not included in the review; discussion of recent flood resilience concepts is largely missing

Page 11 line 285: "an adaptive development planning process" this is increasingly important approach in responding to climate uncertainties and is an area that might be expanded on in further detail.

Page 13 : general comment : What physical interventions were stimulated by this policy evolution?

Page 20 line 468-491 What is the commonality OR uniqueness in the separate approaches described here?

Page 20 line 496 "The cities that are selected in this review rely upon hard engineered defences " – is there space available for other solutions?

Page 22: line 509 " required better development of non-engineering measures". It would be very informative and useful to conduct a Multi Criteria Decision Analysis (MCDA) on alternative strategies, reflecting priorities and weightings that reflect the specific contexts of SE Asian coastal megacities. Such a synthesis that might translate a review of practice elsewhere into practical recommendations for the region would be a potential major contribution the paper could make.

Page 22 line 525 Do the coping strategies referred to relate to individual or institutional level?

Page 22 line 528-529 " The international experience clearly shows that SFRM approach is more complex than control or defend....." but this needs to be qualified with respect to specific local circumstances, contexts and constraints

Page 22 line 539 "....different countries and cities have their own interpretation on SFRM " – reinforces preceding point ( i.e. the importance of context, pointed to by the authors in the concluding paragraph on page 23)

**Proposed corrections**

Page  2 line 43  add  over the *long period* of human history

Page 2 line 44   Add : or a long *duration* precipitation event

Page 3  "2 Learning from the four front-end countries ":  define "front end " unclear what this  is ?

Page 8 line 280  "tropical  cyclones" line 220 Hurricanes : it would be  helpful to precisely distinguish terminology here  clarifying the difference between tropical storms, tropical cyclones, hurricanes and typhoons

Page 8 line 233 replace "Evan"  with "Even"

Page 13 line 357 define "flood resilience"

Page 14 line 375  "Influential to policies", which policies?

Page 15 Figure  4: could this be extended  to include concepts of Urban Flood Resilience

 A useful paper exploring resilience concepts  across wider water management  is:
Elizabeth Lawson, Raziyeh Farmani   Ewan Woodley   and David Butler (2020) A Resilient and Sustainable Water Sector: Barriers to the Operationalisation of Resilience Sustainability **2020**, 12, 1797; doi:10.3390/su12051797
Page 18 line 411: Begin sentence  with : " *In the UK*, local authorities..."

Page 18 line  430 English corrections needed :  " Singapore was pioneered adopted Low Impact Development (LID)...."  (e.g. delete "was pioneered"?)

Page 19 line 445: ...Shanghai during 1981...  -  provide examples  of more recent events ?

Page 19 line 453  replace " favorited" with " preferred"

Page 19 line 454  replace " focusing " with " focussed"

Page 19  line  463: verb required e.g " For example, *the*  Sganghai authority *acted* to raise the flood protection level....).

---

## Community Comment (CC2)

NHESS-2021-268    Submitted on 15 Sep 2021

Comparative analysis and implications of sustainable Flood Risk Management in four front-end countries: The United Kingdom, the Netherlands, the United States, & Japan

Faith Ka Shun Chan, Liang Emlyn Yang, Gordon Mitchell, Nigel Wright, Mingfu Guan, Xiaohui Lu, Zilin Wang, Burrell Montz, and Olalekan Adekola

Special Issue: Future risk and adaptation in coastal cities

Handling Editor: Animesh Gain, again@mit.edu

Title: Response letter of *NHESS-2021-268*

Dear Handling Editor of NHESS, Prof Dr Animesh Gain

On behalf of all co-authors, I would like to appreciate two anonymous reviewers' responses and feedback to our manuscript, namely "Comparative analysis and implications of sustainable Flood Risk Management in four front-end countries: The United Kingdom, the Netherlands, the United States, & Japan" (Ref: NHESS-2021-268) for the journal *NHESS*.

I would like to respond to all suggestions/comments per se as below (start at next page). The reviewers' comments are shown in italics and my responses are shown in blue colour.

I would like to submit the tracked version of our manuscript (Ref: NHESS-2021-268 R1) for the reviewers and editor to read our changes/revisions more explicitly.

We hope this revision will be satisfactory and grateful for the handling operation by the NHESS editorial office, the handling editor, Prof Dr Animesh Gain, and two anonymous reviewers for the feedback and comments of this revision, which is truly appreciated.

Once again, we would like to appreciate all changes and hope our revision has been addressed all issues raised by two reviewers and helped this manuscript to be improved substantially.

Yours sincerely,

Prof Dr Faith Chan

13 Dec 2021

**RC1: 'Comment on nhess-2021-268', Anonymous Referee #1 (R1)**

1. **Specific comments**

R1: *Title: Misleading and does not describe the full purpose of the paper. Unclear what is meant by "front-end cities" and reference to Asian coastal megacities should be made clear*

FC: thanks, and appreciated for the comment and now has revised the title: "Comparison analysis of Sustainable Flood Risk Management by four countries: the United Kingdom, the Netherlands, United States, & Japan and the implications to Asian coastal megacities" in the revised manuscript.

R1: *Page 1: Abstract line 28-30: "This paper reviews the past and present flood management approaches and experiences from flood defence to FRM in four developed countries to highlight lessons for developing mega deltas". The paper does not fully explore the hydrological (and other contextual) dissimilarities between the regions being compared, and whether these dissimilarities can justify and sustain strategies from different parts of the world working elsewhere. Specifically, rapidly "developing mega deltas" bring their own constraints which might be explored in more detail"*

FC: Appreciate the comment and please refer to the revised abstract (line 28-30) in this revised version (see the track changes for our revision). As we changed the sentence to be "*This paper reviews the past and current flood management experiences from flood defence to FRM in four developed countries to highlight lessons for developing coastal megacities.*"

That may be aligned with the article better and thanks for this comment. We will include this in the revised manuscript.

R1: *Page 2 line 37 explain how subsidence arises from human factors (e.g. as a result of excessive groundwater extraction)*

FC: Appreciate the comment and addressed – see line 37-38 and noted "*…to mitigate substantial risks due to human (exist huge populations, rapid socio-economic growth, subsidence from excessive groundwater extraction, etc.)*" We will include this in the revised manuscript.

R1: *Page 2-3 line 84 this focuses on SE Asian examples not reflected in the paper's title (see earlier comment)*

FC: Thanks and the title has been revised. We will include this in the revised manuscript.

R1: *line 86-87 reference is made to hard engineering solutions and flood control. This could be further explained both in respect to the engineering materials used (e.g. concrete) and most importantly the basic driving principle of increasing in- channel conveyance.*
FC: Thanks, and the sentences have been revised (please see lines 86-90). We will include this in the revised manuscript.

R1: *lines 111-112 "These experiences offer lessons from FRM in Asian coastal megacities". Whilst undoubtedly some of the reviewed changes in practice are relevant, such as managing urban flooding through the principle of source control (-no mention-), there are large hydrological (and meteorological) dissimilarities between the areas being compared and this should be explored and acknowledged in more detail.*

FC: Thanks, and the sentences have been revised (please see lines 110-115). We will include this in the revised manuscript and appreciated the suggestion.

R1: *Page 4 lines 120-130 Clearer distinctions could be made between rural and urban flood policy responses. For example, the papers say little about the introduction of Sustainable URBAN drainage (limited to line 153)*

FC: Thanks, we have only illustrated the progress here and have no intention to confuse readers on the rural and urban flood policy response and perspectives. We added, "*In general, the UK Government adopted land drainage and hard-engineered defences such as river straightening, construction of embankments and levees in rural and urban flood policy responses during this era*." (please see lines 120-122). Thanks, and that is helpful. We will include this in the revised manuscript and appreciated the suggestion.

R1: *Page 5 line 135 "be more specific on what is meant by "externality effects"*
FC: Thanks, we have provided the examples of "externality factors", I think using "factors" instead of "effects" is more appropriate, we explain here - such as inflations and market prices of the construction and labour cost, etc. (see line 135-140). That is a helpful suggestion. We will include this in the revised manuscript and appreciated it.

R1: *Pages 16-17 – Tables 2 and 3: information here represents a heavy UK focus*

FC: Thanks, we have no intention to direct readers focusing on UK lessons, but we have provided the evidence and lessons of the definition and principles only and Table 3 also included examples from NL, US and Japan. Thanks for this comment and appreciated it.

R1: *Page 5: general comment: greater and more explicit distinction should be made between pluvial and fluvial flooding (with respect to the strategies considered).*

FC: Thanks, we have addressed these policies according to various flood types and see the revision in pages 5-6. Thanks for this comment and appreciated it. We will include this in the revised manuscript and appreciated it.

R1: *Page 6 line 180 "complex governance" structure; fragmented responsibilities are serious ongoing in issues in UK flood management (e.g. see: Ashley R., Gersonius B., Horton B "Management flooding: from a problem to an opportunity" Royal Society Philosophical Transactions A Volume 378 Issues 2169 April 2020*

FC: Thanks, we have addressed this and please see lines 185-188, we included and cited Ashley et al. 2020 and thanks for the comment and appreciated. We will include this in the revised manuscript and appreciated it.

R1: *Section 2 The most recent references (around 2012-2014) seem somewhat dated with more recent papers on this topic not included in the review; discussion of recent flood resilience concepts is largely missing*
FC: Thanks, we have provided the evidence and facts of the flood management progress during the past decades, but definitely taking this suggestion, we have included the latest progress such as reflected from the Japanese case (e.g. MILT (2018). Thanks for the comment as truly appreciated.

R1: *Page 11 line 285: "an adaptive development planning process" this is increasingly important approach in responding to climate uncertainties and is an area that might be expanded on in further detail.*

FC: Thanks, we have explained the example here, "…such as the implementation of climate adaptation plans merged with the long-term Master plan of the New York City" and see lines 295-298. We will include this in the revised manuscript and thanks for the suggestion and truly appreciated.

R1: *Page 13: general comment: What physical interventions were stimulated by this policy evolution?*

FC: Thanks, we provided the contextual findings of the progress on the FRM in Japan on this page and that is exactly what we want to emphasise that the progress has been developed further from physical to other layers rather than reliance on traditional engineered defences. See lines 360-362. We will include this in the revised manuscript and thanks for the suggestion and truly appreciated.

R1: Page 20 line 468-491 What is the commonality OR uniqueness in the separate approaches described here?

FC: The commonality of Page 20 line 468-491 in the separate approaches described in this section/paragraphs and emphasised the shift of FRM has been transformed considering wider aspects of social-economic risk and health issues of the communities have started to be considered in the FRM policy implementation in this paragraph and that is the commonality (see line 465-466). We will include this in the revised manuscript and thanks for the suggestion and truly appreciated.

R1: Page 20 line 496 "The cities that are selected in this review rely upon hard engineered defences "– is there space available for other solutions?

FC: Thanks, as we selected these cases/coastal cities in Asia are mainly based on their previous FRM approaches on hard-engineered defences and we clarified by adding "…and their previous ways to deal with flooding" (see page 21 line 512). Thanks for this comment. We will include this in the revised manuscript and thanks for the suggestion and truly appreciated.

R1: Page 22: line 509 "required better development of non-engineering measures". It would be very informative and useful to conduct a Multi-Criteria Decision Analysis (MCDA) on alternative strategies, reflecting priorities and weightings that reflect the specific contexts of SE Asian coastal megacities. Such a synthesis that might translate a review of practice elsewhere into practical recommendations for the region would be a potential major contribution the paper could make.

FC: Thanks, as addressed this comment (see page 23 line 527-528) in the revised version. We will include this in the revised manuscript and thanks for the suggestion and truly appreciated.

R1: Page 22 line 525 Do the coping strategies referred to relate to individual or institutional level?

FC: Thanks as we addressed that the coping strategies should be related across individual to institutional levels and see lines 546-547. Appreciated.

R1: Page 22 line 528-529 "The international experience clearly shows that SFRM approach is more complex than control or defend...." but this needs to be qualified with respect to specific local circumstances, contexts and constraints Page 22 line 539 ".... different countries and cities have their own interpretation on SFRM "– reinforces preceding point (i.e. the importance of context, pointed to by the authors in the concluding paragraph on page 23

FC: Thanks for the comment and suggestion as we have emphasised this in the conclusion that should be considered the "local knowledge" for delivering the SFRM. See Lines 560-565 in the revised version. We will include this in the revised manuscript and thanks for the suggestion and truly appreciated.

**2. Proposed corrections**

R1: *Page 2 line 43 add over the long period of human history*
FC: Thanks and following the suggestion and highlighted the changes (see line 43 in page 2). We will include this in the revised manuscript.

R1: *Page 2 line 44 Add: or a long duration precipitation event*
FC: Thanks and following the suggestion and highlighted the changes (see line 44 in page 2). We will include this in the revised manuscript.

R1: *Page 3 "2 Learning from the four front-end countries ": define "front end "unclear what this is? Page 8 line 280 "tropical cyclones" line 220 Hurricanes: it would be helpful to precisely distinguish terminology here clarifying the difference between tropical storms, tropical cyclones, hurricanes and typhoons*
FC: Thanks, as we decided to delete "front end" as just use "4 countries" as addressed in the title as well. Thanks for another comment here for the clarification and explanation of tropical cyclones and difference between tropical storms, cyclones, hurricanes and typhoons. Please see the insertion here from line 230-236 in page 8 (yellow highlighted). We will include the revision in the revised manuscript in the submission.

R1: *Page 8 line 233 replace "Evan" with "Even"*
FC: Thanks, has addressed this and see line 249 in the revised version (yellow highlighted). We will include the revision in the revised manuscript in the submission.

R1: *Page 13 line 357 define "flood resilience"*
FC: Thanks, as we defined the term "flood resilience" and addressed (yellow highlighted) and see line 375-376 page 13. We will include the revision in the revised manuscript in the submission.

R1: *Page 14 line 375 "Influential to policies", which policies?*
FC: thanks, and addressed – that is the "sustainability" policies and highlighted please see line 394 in page 14. We will include the revision in the revised manuscript in the submission.

R1: *Page 15 Figure 4: could this be extended to include concepts of Urban Flood Resilience*

*A useful paper exploring resilience concepts across wider water management is: Elizabeth Lawson, Raziyeh Farmani Ewan Woodley and David Butler (2020) A Resilient and Sustainable Water Sector: Barriers to the Operationalisation of Resilience Sustainability 2020, 12, 1797; doi:10.3390/su12051797*
FC: thanks, and we will try to address this and include the suggested paper into the diagram (Figure 4), otherwise we will provide rebuttal reasons why we could not do this. Thanks for the comment and we will address in the revised manuscript.

R1: *Page 18 line 411: Begin sentence with: "In the UK, local authorities..."*
FC: Thanks and addressed, please see the yellow highlighted (line 411 page 18). We will include this change in the revised manuscript.

R1: *Page 18 line 430 English corrections needed: "Singapore was pioneered adopted Low Impact Development (LID)...." (e.g. delete "was pioneered"?)*
FC: and highlighted (see page 19 line 451 in the revised version). We will include this change Thanks as followed the suggestion in the revised manuscript.

R1: *Page 19 line 445: ...Shanghai during 1981... - provide examples of more recent events?*
FC: Thanks as we provided the recent events and highlighted and see line 467 page 20. Thanks as followed the suggestion in the revised manuscript.

R1: *Page 19 line 453 replace "favorited" with "preferred"*
FC: Thanks as followed the suggestion (see page 20 line 476) in the revised manuscript.

R1: *Page 19 line 454 replace "focusing "with " focussed"*
FC:  Thanks as followed the suggestion (see page 20 line 477) in the revised manuscript.

R1: *Page 19 line 463: verb required e.g "For example, the Sganghai authority acted to raise the flood protection level....)*
FC: Thanks as followed the suggestion (see page 20 line 485) in the revised manuscript.

Reviewer 2 (R2)

Comments:

R2: *This is an interesting and important paper examining the four case studies on how they cope with flooding hopefully with transferrable best practices to Asian Megacities.*

*The paper is detailed and well written but needs to do an overall edit in terms of language and spelling accuracy. There are some terms used in this paper which needs further clarifications and evidence.*

*The methods are sound. Comparing and contrasting the four cases made this a very rich discussion but some sections/ statements need more evidence or clarity.*

*In addition, more could be devoted to the best practices that could be transferred to the Asian context- which is distinct from the chosen cases in terms of geography and political and social structures.*

*Overall a good paper with a bit more work on the transferable practices and overall edit in terms of clarity and language would be make this an excellent paper to be published in the journal.*

FC: Thanks for the positive comments and truly appreciated, we will address all grammatical and provide overall edits in terms of clarity and improve the language issue to make our manuscript to be better in the revised version. Thanks for the comment and truly appreciated it.

---

## Author Response (AR1)

NHESS-2021-268

"Comparative analysis and implications of sustainable Flood Risk Management in four front-end countries: The United Kingdom, the Netherlands, the United States, & Japan"

Faith Ka Shun Chan, Liang Emlyn Yang, Gordon Mitchell, Nigel Wright, Mingfu Guan, Xiaohui Lu, Zilin Wang, Burrell Montz, and Olalekan Adekola

Special Issue: Future risk and adaptation in coastal cities

Handling Editor: Animesh Gain, again@mit.edu

Title: Response letter of *NHESS-2021-268* R1

Dear Handling Editor of NHESS, Prof Dr Animesh Gain

On behalf of all co-authors, I would like to appreciate two anonymous reviewers' responses and feedback to our manuscript, namely "Comparative analysis and implications of sustainable Flood Risk Management in four front-end countries: The United Kingdom, the Netherlands, the United States, & Japan" (Ref: NHESS-2021-268) for the journal *NHESS*.

I would like to respond to all suggestions/comments per se as below (start at next page). The reviewers' comments are shown in italics and my responses are shown in blue colour.

I would like to submit the tracked version of our manuscript (Ref: NHESS-2021-268 R1), and revise the title as" *Comparison of Sustainable Flood Risk Management by four countries: the United Kingdom, the Netherlands, the United States, & Japan, and the implications for Asian coastal megacities*" for the reviewers and editor to read our changes/revisions more explicitly.

We hope this revision will be satisfactory and grateful for the handling operation by the NHESS editorial office, the handling editor, Prof Dr Animesh Gain, and two anonymous reviewers for the feedback and comments of this revision, we have tried our best efforts to address all issues in the manuscript. All comments and suggestions have been greatly improved the quality of the manuscript, which is truly appreciated.

Once again, we would like to appreciate all changes and hope our revision has been addressed all issues raised by two reviewers and helped this manuscript to be improved substantially.

Yours sincerely,

Dr Faith Chan

6 Jan 2022

**RC1: 'Comment on *nhess*-2021-268', Anonymous Referee #1 (R1)**

**1.       Specific comments**

R1: *Title: Misleading and does not describe the full purpose of the paper. Unclear what is meant by "front-end cities" and reference to Asian coastal megacities should be made clear*

FC: Thanks, and appreciated for the comment and now has revised the title: "Comparison analysis of Sustainable Flood Risk Management by four countries: the United Kingdom, the Netherlands, United States, & Japan and the implications for Asian coastal megacities" in the revised manuscript.

 R1: *Page 1: Abstract line 28-30: "This paper reviews the past and present flood management approaches and experiences from flood defence to FRM in four developed countries to highlight lessons for developing mega deltas". The paper does not fully explore the hydrological (and other contextual) dissimilarities between the regions being compared, and whether these dissimilarities can justify and sustain strategies from different parts of the world working elsewhere. Specifically, rapidly "developing mega deltas" bring their constraints which might be explored in more detail"*

FC: Appreciate the comment and please refer to the revised abstract (see line 28-30) in this revised version (see the track changes for our revision). As we changed the sentence to be "*This paper reviews the past and current flood management experiences from flood defence to FRM in four developed countries to highlight lessons for coastal megacities in development*."

That may be aligned with the article better and thanks for this comment. We include this in the revised manuscript.

R1: *Page 2 line 37 explain how subsidence arises from human factors (e.g. as a result of excessive groundwater extraction)*

FC: Appreciate the comment and addressed – see lines 37-38 and noted "*…to mitigate substantial risks due to human factors (exist huge populations, rapid socio-economic growth, subsidence from excessive groundwater extraction, etc.) …*" We will include this in the revised manuscript.

R1: *Page 2-3 line 84 this focuses on SE Asian examples not reflected in the paper's title (see earlier comment)*

FC: Thanks and the title has been revised. We addressed this issue in the revised manuscript.

R1*: line 86-87 references is made to hard engineering solutions and flood control. This could be further explained both in respect to the engineering materials used (e.g. concrete) and most importantly the basic driving principle of increasing in- channel conveyance.*
FC: Thanks and the sentences have been revised (please see lines 86-90). We addressed this comment and see the changes in the revised manuscript.

R1: *lines 111-112 "These experiences offer lessons from FRM in Asian coastal megacities". Whilst undoubtedly some of the reviewed changes in practice are relevant, such as managing urban flooding through the principle of source control (-no mention-), there are large hydrological (and meteorological) dissimilarities between the areas being compared and this should be explored and acknowledged in more detail.*

FC: Thanks and the sentences have been revised (please see lines 110-115). We addressed this comment and see the changes in the revised manuscript.

R1: *Page 4 lines 120-130 Clearer distinctions could be made between rural and urban flood policy responses. For example, the papers say little about the introduction of Sustainable URBAN drainage (limited to line 153)*

FC: Thanks, we have only illustrated the progress here and have no intention to confuse readers on the rural and urban flood policy response and perspectives. We added, "*In general, the UK Government adopted land drainage and hard-engineered defences such as river straightening, construction of embankments and levees in rural and urban flood policy responses during this era.*" (please see lines 120-122). Thanks, and that is helpful. We include this in the revised manuscript and appreciated the suggestion.

R1: *Page 5 line 135 "be more specific on what is meant by "externality effects"*
FC: Thanks, we have provided the examples of "externality factors", I think using "factors" instead of "effects" is more appropriate, we explain here - such as inflations and market prices of the construction and labour cost, etc. (see line 140-141 in the tracked version). That is a helpful suggestion. We addressed this issue in the revised manuscript and appreciated it.

R1: *Pages 16-17 – Tables 2 and 3: information here represents a heavy UK focus*

FC: Thanks, we have no intention to direct readers focusing on UK lessons, but we have provided the evidence and lessons of the definition and principles only and Table 3 also included examples from NL, US and Japan. Thanks for this comment and appreciated it.

R1: *Page 5: general comment: greater and more explicit distinction should be made between pluvial and fluvial flooding (with respect to the strategies considered).*

FC: Thanks, we have addressed these policies according to various flood types and see the revision on pages 5-6. Thanks for this comment and appreciated it.

R1: *Page 6 line 180 "complex governance" structure; fragmented responsibilities are serious ongoing in issues in UK flood management (e.g. see: Ashley R., Gersonius B., Horton B "Management flooding: from a problem to an opportunity" Royal Society Philosophical Transactions A Volume 378 Issues 2169 April 2020*

FC: Thanks, we have addressed this and please see lines 185-188, we included and cited Ashley et al. 2020 and thanks for the comment and appreciated.

R1: *Section 2 The most recent references (around 2012-2014) seem somewhat dated with more recent papers on this topic not included in the review; discussion of recent flood resilience concepts is largely missing*
FC: Thanks, we have provided the evidence and facts of the flood management progress during the past decades, but definitely taking this suggestion, we have included the latest progress such as reflected from the Japanese case (e.g. MILT (2018). Thanks for the comment as truly appreciated.

R1: *Page 11 line 285: "an adaptive development planning process" this is increasingly important approach in responding to climate uncertainties and is an area that might be expanded on in further detail.*

FC: Thanks, we have explained the example here, "…such as the implementation of climate adaptation plans merged with the long-term Master plan of the New York City" and see lines 305-307. We addressed this issue in the revised manuscript and thanks for the suggestion and truly appreciated.

R1: *Page 13: general comment: What physical interventions were stimulated by this policy evolution?*

FC: Thanks, we provided the contextual findings of the progress on the FRM in Japan on this page and that is exactly what we want to emphasise that the progress has been developed further from physical to other layers rather than reliance on traditional engineered defences. See lines 366-372. We addressed this issue in the revised manuscript and thanks for the suggestion and truly appreciated.

R1: Page 20 line 468-491 What is the commonality OR uniqueness in the separate approaches described here?

FC: The commonality of Page 20 line 468-491 (in last version), which is the separate approaches as described in this section/paragraphs and emphasised the shift of FRM has been transformed considering wider aspects of social-economic risk and health issues of the communities have started to be considered in the FRM policy implementation in this paragraph and that is the commonality (see page 20-22). We addressed this comment in in the revised manuscript and thanks for the suggestion and truly appreciated.

R1: Page 20 line 496 "The cities that are selected in this review rely upon hard engineered defences "– is there space available for other solutions?

FC: Thanks, as we selected these cases/coastal cities in Asia are mainly based on their previous FRM approaches on hard-engineered defences and we clarified by adding "…and their previous ways to deal with flooding" (see page 22 line 530). Thanks for this comment. We addressed this in the revised manuscript and thanks for the suggestion and truly appreciated.

R1: Page 22: line 509 "required better development of non-engineering measures". It would be very informative and useful to conduct a Multi-Criteria Decision Analysis (MCDA) on alternative strategies, reflecting priorities and weightings that reflect the specific contexts of SE Asian coastal megacities. Such a synthesis that might translate a review of practice elsewhere into practical recommendations for the region would be a potential major contribution the paper could make.

FC: Thanks, as addressed this comment (see page 25 line 550-552) in the revised version. We addressed this in the revised manuscript and thanks for the suggestion and truly appreciated.

R1: Page 22 line 525 Do the coping strategies referred to relate to individual or institutional level?

FC: Thanks as we addressed that the coping strategies should be related across individual to institutional levels and see lines 580-587. Appreciated.

R1: Page 22 line 528-529 "The international experience clearly shows that SFRM approach is more complex than control or defend...." but this needs to be qualified with respect to specific local circumstances, contexts and constraints Page 22 line 539 ".... different countries and cities have their own interpretation on SFRM "– reinforces preceding point (i.e. the importance of context, pointed to by the authors in the concluding paragraph on page 23

FC: Thanks for the comment and suggestion as we have emphasised this in the conclusion that should be considered the "local knowledge" for delivering the SFRM. See Lines 580-585 on page 26 (revised version) in the concluding paragraph. We addressed this in the revised manuscript and thanks for the suggestion and truly appreciated.

**2.       Proposed corrections**

R1: *Page 2 line 43 add over the long period of human history*
FC: Thanks and following the suggestion and highlighted in yellow for the changes (see line 44 in page 2).

R1: *Page 2 line 44 Add: or a long duration precipitation event*
FC: Thanks and following the suggestion and highlighted in yellow for the changes (see line 45 in page 2).

R1: *Page 3 "2 Learning from the four front-end countries ": define "front end "unclear what this is? Page 8 line 280 "tropical cyclones" line 220 Hurricanes: it would be helpful to precisely distinguish terminology here clarifying the difference between tropical storms, tropical cyclones, hurricanes and typhoons*
FC: Thanks, as we decided to delete "front end" as just use "4 countries" as addressed in the title as well. Thanks for another comment here for the clarification and explanation of tropical cyclones and difference between tropical storms, cyclones, hurricanes and typhoons. Please see the insertion here from line 234-242 in page 8 (yellow highlighted). We addressed this in the revised manuscript.

R1: *Page 8 line 233 replace "Evan" with "Even"*
FC: Thanks, has addressed this and see line 255 in the revised version (yellow highlighted). We addressed this in the revised manuscript.

R1: *Page 13 line 357 define "flood resilience"*
FC: Thanks, as we defined the term "flood resilience" and addressed (yellow highlighted) and see line 381-382 page 13. We addressed this in the revised manuscript.

R1: *Page 14 line 375 "Influential to policies", which policies?*
FC: Thanks, and addressed – that is the "sustainability" policies and highlighted please see line 400 in page 14. We addressed this in the revision in the revised manuscript in the submission.

R1: *Page 15 Figure 4: could this be extended to include concepts of Urban Flood Resilience*

*A useful paper exploring resilience concepts across wider water management is: Elizabeth Lawson, Raziyeh Farmani Ewan Woodley and David Butler (2020) A Resilient and Sustainable Water Sector: Barriers to the Operationalisation of Resilience Sustainability 2020, 12, 1797; doi:10.3390/su12051797*
FC: Thanks, and we appreciated for the comment but the figure is definitely implied the progress "towards" SFRM and not reflect the existing paradigm of urban flood resilience, the figure has indicated the progress on the stages and progress of flood management and that is our purpose, we appreciate the suggestion, but we prefer to keep the figure as it is. Thanks for the comment.

R1: *Page 15 line 411: Begin sentence with: "In the UK, local authorities..."*
FC: Thanks and addressed, please see the yellow highlighted (line 416-417 page 15). We addressed this in the revision in the revised manuscript in the submission.

R1: *Page 18 line 430 English corrections needed: "Singapore was pioneered adopted Low Impact Development (LID)...." (e.g. delete "was pioneered"?)*
FC: Thanks as highlighted (see page 19 line 463 in the revised version). We addressed this in the revision in the revised manuscript in the submission.

R1: *Page 19 line 445: ...Shanghai during 1981... - provide examples of more recent events?*
FC: Thanks as we provided the recent events and highlighted and see line 478-479 page 21. Thanks as followed the suggestion in the revised manuscript.

R1: *Page 19 line 453 replace "favorited" with "preferred"*
FC: Thanks as followed the suggestion (see page 21 line 487) in the revised manuscript.

R1: *Page 19 line 454 replace "focusing "with " focussed"*

FC:  Thanks as followed the suggestion (see page 21 line 488) in the revised manuscript.

R1: *Page 19 line 463: verb required e.g "For example, the Sganghai authority acted to raise the flood protection level....)*

FC: Thanks as followed the suggestion (see page 21 line 498) in the revised manuscript.

Reviewer 2 (R2)

Comments:

R2: *This is an interesting and important paper examining the four case studies on how they cope with flooding hopefully with transferrable best practices to Asian Megacities.*

*The paper is detailed and well written but needs to do an overall edit in terms of language and spelling accuracy. There are some terms used in this paper which needs further clarifications and evidence.*

*The methods are sound. Comparing and contrasting the four cases made this a very rich discussion but some sections/ statements need more evidence or clarity.*

*In addition, more could be devoted to the best practices that could be transferred to the Asian context- which is distinct from the chosen cases in terms of geography and political and social structures.*

*Overall a good paper with a bit more work on the transferable practices and overall edit in terms of clarity and language would be make this an excellent paper to be published in the journal.*

FC: Thanks for the positive comments and truly appreciated, we addressed all grammatical and provide overall edits in terms of clarity and improve the language issue to make our manuscript to be better in this revised version. Thanks for the comment and truly appreciated it.

---

## Author Response (AR2)

**NHESS-2021-268R2**

**"Comparative analysis and implications of sustainable Flood Risk Management in four front-end countries: The United Kingdom, the Netherlands, the United States, & Japan"**

Faith Ka Shun Chan, Liang Emlyn Yang, Gordon Mitchell, Nigel Wright, Mingfu Guan, Xiaohui Lu, Zilin Wang, Burrell Montz, and Olalekan Adekola

Special Issue: Future risk and adaptation in coastal cities

Handling Editor: Animesh Gain, again@mit.edu

**Title: Response letter of *NHESS-2021-268* R2**

Dear Handling Editor of NHESS, Prof Dr Animesh Gain

On behalf of all co-authors, I would like to appreciate two anonymous reviewers' responses and feedback to our manuscript, namely "Comparative analysis and implications of sustainable Flood Risk Management in four front-end countries: The United Kingdom, the Netherlands, the United States, & Japan" (Ref: NHESS-2021-268R1) for the journal *NHESS*.

I would like to respond to all suggestions/comments per se as below (start on the next page). The reviewers' comments are shown in italics and my responses are shown in blue colour.

I would like to submit the tracked version of our manuscript (Ref: NHESS-2021-268 R2_tracked), and revise the title to" *Comparison of Sustainable Flood Risk Management by four countries: the United Kingdom, the Netherlands, the United States, & Japan, and the implications for Asian coastal megacities*" for the reviewers and editor to read our changes/revisions more explicitly.

We hope this revision will be satisfactory and grateful for the handling operation by the NHESS editorial office, the handling editor, Prof Dr Animesh Gain, and two anonymous reviewers for the feedback and comments on this revision, we have tried our best efforts to address all issues in the manuscript. All comments and suggestions have greatly improved the quality of the manuscript, which is truly appreciated. The authors have appreciated the efforts of you and both reviewers and are delighted both reviewers' comments are positive and helpful, and reviewer 1 provided the acceptance of this manuscript and reviewer 3 provided a minor revision alongside some constructive comments to improve the manuscript further at this stage. On behalf of all co-authors, we are truly appreciated this valuable feedback and comments from you and two anonymous reviewers.

Once again, we would like to appreciate all changes and hope our revision has addressed all issues raised by two reviewers and helped this manuscript to be improved further.

Yours sincerely,

Dr Faith Chan

10 May 2022

**'Comment on *nhess*-2021-268R1', Anonymous Referee #3**

**General**

R3: *"Interesting description of the development in the flood risk strategies/policies in the 4 countries, and later you present an interesting overview of strategies in Asian cities, underpinned with an impressive list of references. However, the topics seem not well-connected (due to the structure of the paper).*

*And you miss a few important developments, e.g. why did you not mention the EU Flood Risk Directive (that was implemented after the 2002 Elbe floods) other than in Table 3? Then you could have compared that with the shift in approach in the US (L. 271).*

*Although you mention in the title and the introduction, that the comparison of the 4 countries is meant to improve urban flood risk management in Asian coastal megacities, there is in your analysis a lot of attention on the general flood risk management policies, and there is (in particular in your description of the UK and Dutch approach) quite a lot of focus on the rural area, instead of on the urban area. In the Netherlands, there are interesting examples of Room for the River for the urban area (e.g. Nijmegen).*

*Table 4 Development of flood management practices in selected E. and S.E. Asian cities, presents a very nice overview."*

FC: Thanks, and appreciated the comment we have addressed these issues – see Table 3 and mentioned the EU Flood Risk Directive and compared it with the shift in approach in the US in the Table (see p.18 in the tracked version) and grateful for this comment.

Thanks for your further comments as this article mainly is using the 4 countries and taking according to lessons for improving the urban flood risk management in Asian coastal megacities, therefore we have put analyses by these countries' latest practices. Our focus is not only on urban or rural areas, actually, but that is also covering the practices of the countries themselves at the national level of their flood risk management practices. But we appreciated the suggestion and mentioned the Nijmegen land management plan in Table 1 (see tracked version p.7).

Thanks for the comments and appreciated the comment that we are pleased the reviewer is happy about table 4.

**1.      Specific comments**

**R3: *Introduction***

*L.44-45: mention here already the different types of floods and what is causing them: coastal flooding: storm, high tides, SLR and not sufficient protection; river/fluvial floods: snowmelt or high precipitation in catchment area leading to flash floods or riverine floods; pluvial floods: extreme rainfall and failing drainage systems.*

FC: Thanks, and have revised it by following the advice (see L45-48 in the tracked version).

*L.48: not the flood hazard is increasing, but the risk, the frequency, or impact of floods is increasing*

FC: Thanks, as revised (see L48 in the tracked version)

*L.50: also soil subsidence (e.g. Jakarta)*

FC: Thanks, as revised (see L51 in the tracked version) as we have also mentioned Jakarta in L56 too.

*L.54: There are actually plans to relocate Jakarta*

FC: Thanks, and appreciated the comment revised (see L54-55 in the track version).

**R3: Learning from the four countries**

*L.88: in low lying deltaic areas there are for centuries flood defences to protect villages and towns (e.g. in China, Netherlands, Germany, US). Please mention, and refer to literature on this.*

FC:  Thanks, as revised and cited the relevant lit works on this e.g. Wang et al. (2013); Van Stokkom et al. (2015); Parker and Fordham (1996) and Arnell (1984) and add a sentence on this (see L89-91 in the track version).

*L.109: here you mention that you start with the analysis of the flood risk strategies in Asian cities, and then next (L.110) on UK, the Netherlands, the US, and Japan. The title of your article suggest the other way around.*

FC: thanks, as revised and we purposed to consider the flood management experiences in the UK, the Netherlands, the US and Japan as noted in the title. Afterwards, these valuable experiences offer lessons for FRM in Asian coastal megacities (see Line 115-120 in the tracked version).

*L.123: do you have indications if this floods and damage in 1947 and 1953 would not have happened with another flood risk strategy? This approach prevented many other (minor) floods! Please rewrite this paragraph.*

FC: thanks, as revised the paragraph and emphasised that the land drainage and hard-engineering approach has successfully protected minor floods and protected farms and agricultural premises during that era as rewritten the paragraph (see lines 127-131 in the tracked version).

*L.127 Sugar beets*

FC: thanks, and added sugar beets (see line 135 in the tracked version).

*L.193 _ 203-205: this is outdated information. There is a new risk based approach implemented (see e.g. Jorissen et al. 2016. Dutch flood protection policy and measures based on risk assessment)*

FC: Thanks very much for this comment and citation as I have revised the paragraph and included these findings (see Lines 235-242 in the tracked version).

*L.198: please also refer to the summer floods of 2021 (due to extreme rainfall).*

FC: thanks, and included the summer floods and impacts of 2021 (see lines 209-214 in the tracked version).

*L. 207: please refer to your used source for the costs (and check!) of the 1995 evacuation.*

FC: thanks, and checked and revised the figures and economic impacts for these two floods (see lines 236-242 in the tracked version) and appreciated the suggestion.

*L. 208: (Olivier and Wytze, 2006) this does not seem like a correct reference*

FC: agree and deleted this reference.

*Table 1: last line: Germany (Deutschland is in German language)*

FC: thanks, and revised (see table 1 in the track version with this revision)

*L. 231: Why do divide between floods and floods due to hurricanes? In addition to coastal floods, hurricanes can also result in riverine floods because extreme precipitation (e.g. Harvey)? (actually, you also raise that in L. 240).*

FC: thanks, and adjusted this issue, also revised the sentences and addressed that is not a divine but the combined effect to enhance the "compound flood", and enhance the riverine flood from extreme rainstorms (see L257-258 in the track version in this revision).

*L. 238-240: not needed to explain this so extensive (but rather mention 'hurricanes (named cyclones or typhoons in other parts of the world)' and refer to documents that explain this).*

FC: thanks, and followed the instruction that deleted "…*but is only used for intense low-pressure weather systems in the NW Pacific*) and just cited the Bureau of Meteorology (2021) (see Lines 255-257 in the tracked version).

*L. 250-254: add a reference.*

FC: Thanks, and added a reference to Arnell (1984) (see line 274 in the revised track version).

*L. 267: MSW is the UK policy, while Room for the River is the Dutch policy.*

FC: Thanks, and addressed (see lines 288-289 in the revised track version).

*L. 291: Also the UK has introduced insurance as an flood risk approach, and as a planning instrument. You could add this to your description of the UK approach*

FC: Thanks for this comment as this is a very good suggestion, we have revised and added the UK flood insurance practice in this paragraph/sub-section (see lines 324-327 in the revised tracked version).

*L. 297: please add reference.*

FC: thanks, and added Bagstad et al. (2007) (see line 316 in the revised tracked version)

*L. 302: word 'also' is missing.*

FC: thanks, and added "also" to that sentence – "…*but also to cost-sharing, the sacrifice of very high-risk areas*, …" (see line 323 in the revised tracked version)

**Discussion**

*L. 417: refer to EU Flood Risk Directive.*

FC: thanks, revised and cited the EU Flood Directive – "*Directive 2007/60/EC of the European Parliament and of the Council of 23 October 2007 on the assessment and management of flood risks (Text with EEA relevance)*" see lines 441-443 in the tracked version.

*L. 425: Table 2?*

FC: thanks, and addressed (see the revised version at line 451).

*Table 2: Here you introduce something completely new, namely the differences within a country. What does it add to your message and approach (the comparison between 4 countries)? I suggest to remove Table 2.*

FC: yes, that is a good suggestion, we removed table 2 as taken the comment – appreciated that. That should be improved the clarity of the sub-section.

*Table 3: is not well structured, and very confusing. In column 1 is very unbalanced. Row 1: in the last column you also mention US (next to UK). As a start for the restructuring process, you should better consider what information you want to present in order to underpin your message.*

FC: thanks, as taking the advice from the last comment, now addressed table 3 now has been revised as table 2.

Also, we have improved the table (i.e. Column 1 and Row 1) and expressed the message clearly from the table (now table 2) we want to cover the major issues and the strengthens and weaknesses of current FRM practices towards SFRM practices across 4 countries and that is the reason we put the examples cover these countries of UK, NL, US and JP and appreciated this comment.

*L. 451-458: a good (and well referenced) overview of Asian cities, however, you do not explain why you start to focus on Singapore.*

FC: Thanks for the advice and appreciation - as I have explained the reason why starts with Singapore – *"Looking at the past (before the 1990s), there are many coastal cities that suffered from severe floods in Asia. Taking Singapore as an example, Singapore has developed rapidly with tremendous socio-economic growth, but has been inundated by severe floods because of the rapid urbanisation since its independence in 1965 (Chan et al., 2018)."*

*L459 and further: This section with examples is interesting, but quite anecdotic. Explaining the structure (e.g. based on Table 4) might improve this section.*

FC: Thanks for this comment, as we have further explained the structure based on table 4 in this section – *"In table 4, we illustrate the flood management practice in Asian cities relative to the dominant flood paradigm in the West, as we discuss the progress of their flood management strategies as we start with the era before the 1990s – "Flood protection (control and defence)"; and further discuss their progress in the 2000s "Flood risk management (FRM)" era and "Post-2000s Sustainable flood risk management (SFRM)" era, we also discuss further for the evidence and examples for these practices among the selected Asian coastal cities further below in this section."* See Lines 477-487 in the revised track version.

*L.480: did the engineering solutions stop after the 1990s?*

FC: thanks, no, the engineering solutions were only particularly popular in that era as we emphasised that – thanks (see line 510 in the revised track version).

*L.491: what do you mean with socio-economic implementation?*

FC: thanks, and clarified (see line 520 in the tracked version).

*L.526-527: there a plenty of more recent references.*

FC: thanks, and added some latest related citations (see lines 557-558).

*L.530-533: please mention the level of protection (e.g. return times) provided by the other approaches.*

FC: yes, thanks for this comment and addressed (see line 562 in the tracked version).

*Table 4 presents a nice overview (with references) and could be compared with Figure 4 (which is actually a Table). However, Figure 4 is on policies of countries, while Table 4 is on cities.*

FC: Thanks for the comment, now revised version table 4 is now revised as table 3 in this version (see p 24 in the tracked version). Thanks for the comment we added the country besides the cities we discussed as the main purpose of figure 4 is to show the progress and lessons for the FRM that reflected or might reflect the Asian coastal cities in Table 3. Thanks for this comment.

*L. 545: here you state that the increasing costs urge for more integrated solutions, but earlier you stated that it was because engineering works could not provide enough safety.*

FC: Thanks for the comment, we are meaning the need for implementing the SFRM practice because the costs of traditional engineered flood defences are increasingly unaffordable, "*A key impetus of the shift in practice from the flood protection and defence paradigms, to SFRM, has been a recognition that the costs of traditional hard engineered flood defences are increasingly unaffordable, and that a wider package of measures is needed to address flood risk.*" Appreciate the comments.

**Conclusion**

*L.576: You described the policies in 4 countries as case studies. Not case studies in 4 countries.*

FC: thanks, and revised – appreciated and see Line 607 in the tracked version.

*L. 584: Your examples of 'Making space for Water' and 'Room for the River' are in rural area. You did not specific present examples or urban area in the 4 countries. So, it not really comparable.*

FC: thanks, as addressed we are not trying to compare these concepts in urban or rural areas but apply to both. See the revised version in lines 615 to 619.